# Effects of Equine-Assisted Activities and Therapies for Individuals with Autism Spectrum Disorder: Systematic Review and Meta-Analysis

**DOI:** 10.3390/ijerph20032630

**Published:** 2023-02-01

**Authors:** Ningkun Xiao, Khyber Shinwari, Sergey Kiselev, Xinlin Huang, Baoheng Li, Jingjing Qi

**Affiliations:** 1Department of Psychology, Institution of Humanities, Ural Federal University, Yekaterinburg 620075, Russia; 2Department of Immunochemistry, Institution of Chemical Engineering, Ural Federal University, Yekaterinburg 620075, Russia; 3Engineering School of Information Technologies, Telecommunications and Control System, Ural Federal University, Yekaterinburg 620002, Russia

**Keywords:** equine-assisted activities and therapies, autism spectrum disorder, social and behavior function, family function

## Abstract

Autism spectrum disorder (ASD) has become a critical public health issue that affects more than 78 million people. In many recent studies, the authors have demonstrated that equine-assisted activities and therapies (EAATs) can substantially improve the social and behavioral skills of children with ASD. However, the qualities of the studies differ, and some authors reached opposite conclusions. In this review, we systematically and objectively examined the effectiveness of EAATs for people with ASD, combining both qualitative and quantitative methods. We searched five databases (PubMed, Scopus, ERIC, ProQuest, and MEDLINE) and added relevant references, and we identified 25 articles for data extraction and analysis. According to our results, EAAT programs can substantially improve the social and behavioral functioning and language abilities of children with ASD. However, among the subdomains, the results were inconsistent. According to the meta-analyses, there were considerable improvements in the social cognition, communication, irritability, and hyperactivity domains, but not in the domains of social awareness, mannerisms, motivation, lethargy, stereotypy, or inappropriate speech. Moreover, there was a lack of sufficient comparative data to conclude that EAAT programs lead to substantial improvements in motor and sensory functioning. In addition, among the included studies, we noted the indicator of whether EAAT programs decreased parental stress and improved family functioning, and although there were four articles in which the researchers considered this aspect, we were unable to draw any conclusions because of the insufficient data and conflicting descriptive evidence. However, we need to consider the improvement in parental mental health as a factor in the effectiveness of this complementary intervention. We hope that in future studies, researchers will focus on family functioning and conduct more randomized controlled trials (RCTs) with blinded assessments using different scales and measures.

## 1. Introduction

### 1.1. ASD

Over the past half-century, the prevalence of ASD has dramatically increased [1]. Currently, ASD affects more than 78 million people worldwide, and it has become a critical public health issue [2]. ASD is a restricted, lifelong, innate, and complex neurodevelopmental disorder that hampers social interactions, cognitive functioning, and perceptual abilities, and it has a high incidence of associated mental retardation issues that can even cause individuals to die by suicide [3,4]. Individuals with ASD frequently exhibit a clinically heterogeneous and considerable proportion of uncontrollable repetitive patterns, emotional dysfunction, and reduced verbal and nonverbal communication during interactions, including less eye contact and body language [5]. Thus, people with ASD are more likely to struggle with multiple communicative and cognitive comorbidities that prevent them from strengthening relationships with others, which results in detrimental social relationships when compared with those of their typically developing peers or people with other psychopathologies [6,7].

Meanwhile, due to the difficulty of self-managing emotions, impaired self-regulation is a predisposition that is inherent in ASD [8] and that can lead to individuals with ASD experiencing more emotion-related problems in daily life than their non-autistic peers (e.g., symptoms of anxiety, depression, aggression, irritability, hyperactivity, rule-breaking, elopement, sensory processing, and sleep) [9,10,11]. Furthermore, individuals with ASD tend to be less adept at employing emotional strategies to self-manage emotion due to their difficulties forming and maintaining friendships, poor academic performances, and participation in social activities, which include experiences such as bullying and exclusion [12,13,14]. These core symptoms frequently persist after childhood and co-occur in adolescence and adulthood [15]. In particular, in a recent study, the authors revealed that there is an increased probability of additional co-occurring symptoms in children with autism, most typically attention deficit hyperactivity disorder (ADHD), oppositional defiant disorder (ODD), and anxiety [16].

The difficulties that autistic people present also impact family members, particularly parents. According to a 2012 study, the parents of autistic children face more mental health difficulties than other parents [17]. Although parent perspectives are critical in understanding the impacts and processes of ASD interventions [18], parents struggle to choose from the many treatment options [19], which complicates the care for people with ASD.

### 1.2. Therapies for ASD

Therefore, the development of an effective treatment has been given priority in this area for individuals with ASD. In recent years, some researchers have shown that the genetic, immunological, environmental, and epigenetic factors are the most important factors in the development and progression of ASD [20]. Researchers try to look for effective and efficient approaches to treating ASD based on pathophysiology and syndromes, typically focusing on six primary areas: (1) sensory integration and sensory-based interventions; (2) relationship-based interactive interventions; (3) skill-based developmental programs; (4) social cognitive skills training; (5) parent-directed or parent-mediated approaches; (6) intensive behavioral interventions [2,21]. Following decades of research, conventional interventions have increasingly become concerned with fostering a positive environment for social engagement and self-regulation in people with ASD, attenuating the negative effects of autistic traits, and enhancing the quality of life and wellbeing in the ASD population [22]. Meanwhile, since ASD is a multifactorial disease, numerous treatment options have become available, and complementary and alternative medicine (CAM) may increasingly be used alongside classical medical practices to treat ASD [23,24,25]. Among the most popular and successful forms of CAM were swimming, music therapy, art therapy, and animal-assisted interventions (AAIs) [25].

### 1.3. Animal-Assisted Interventions (AAIs) for ASD

Under the influence of these studies, AAIs have entered the field of vision of researchers, and it is one of the most promising areas for remediating the core impairments of people with ASD [26,27]. AAIs include a variety of animals, such as dogs, horses, rabbits, dolphins, guinea pigs, and llamas, and incorporating animals into therapeutic treatments appears to effectively decrease the problematic behaviors and improve social communication for ASD populations [28]. In numerous emerging studies in recent years, researchers have laid the groundwork for the use of AAIs to assist individuals with ASD in regulating their emotions, improving their cognitive domains and social communication functioning, engaging in prosocial actions, and reducing maladaptive repetitive behaviors that are associated with stress [28,29,30,31,32,33].

### 1.4. EAATs for ASD

EAATs can help people with autism, cerebral palsy, intellectual disabilities, multiple sclerosis (MS), and post-traumatic stress disorder (PTSD), among other conditions [34,35]. Of all the animal-assisted therapies for ASD, the EAAT program is the most widely utilized [36]. In a 2018 study [37], the authors found that 10% of the parents of children with ASD have used therapies or interventions that include horses, which could be because EAATs have benefits that are different from those of other animal-assisted therapies. In three studies [38,39,40], the authors noted that the rhythmic movements of horseback riding can especially activate the vestibular systems of children with ASD, which can enhance their speech production and improve their learning outcomes. Riders have to actively manage their own body behaviors, which promotes their capacities for voluntary control and nonverbal communication. Another analysis revealed that the hippotherapy (HIP) exercises had a beneficial effect on postural control, interpersonal relationships, and adaptive behaviors [41]. Therefore, horses can offer people with ASD a special way of fostering positive social engagement [32]. Other researchers [28,40,42,43] have offered another insight: that the effects of horseback riding interventions might be optimally shaped by the relationship that forms over time between all humans and horses in groups, including a series of training steps and an accumulation of stimuli to elicit social interaction. From the same perspective, occupational therapists frequently employ “catalyst techniques” to increase arousal emotions, which contributes to behavioral and multisensory perception improvements [28,44,45]. In summary, because these interventions created upbeat and happy environments and provided multisensory stimulation, the benefits of horseback riding for people with ASD were enhanced. Equine-assisted interventions (EAIs) for ASD are rapidly increasing as a complementary therapy for ASD [32,36].

EAIs are programs that incorporate horses to provide rehabilitative and educational benefits to the participants [46]. EAIs are typically referred to as EAATs, and they include two main types of interventions: (1) equine-assisted therapies (EATs), which include hippotherapy (HIP) and equine-assisted psychotherapy (EAP), and equine-assisted activities (EAAs), which include therapeutic horseback riding (THR) [42]. Each method has a different specific therapeutic focus [47]. The fundamental and core idea behind THR is to engage people with ASD in horseback riding and nonriding activities (Barn activities, such as cleaning the barn, feeding horses, and watching the horses’ motions) with licensed instructors, counselors, or equestrians who teach them horsemanship skills that target improving their physical, behavioral, and prosocial health [38]. Hippotherapy utilizes occupational or physical therapy exercises by using the horse’s movements to improve the engagement of the sensory, neuromotor, and cognitive systems to improve the functional outcomes. Equine-assisted occupational therapy (EAOT), therapeutic riding (TR), and equine-facilitated learning (EFL) are a few of the other common terms. Notably, as opposed to EATs and HIP, equine-facilitated learning (EFL) is a distinct experiential learning technique that blends learning abilities and interaction with horses (ponies, miniature horses, donkeys, and mules) with individual therapy and emotional regulation to strengthen children’s awareness and control of their emotions, cognition, and behavior [48].

### 1.5. Relevance

In numerous studies, researchers have demonstrated that the participation of autistic people in therapeutic horseback riding programs improves their social interaction, socialization, and stereotyping behaviors. However, these therapies and interventions still need to be improved [29,49], and the extent to which these improvements occur, as well as whether the involvement duration and type of treatment have an impact on the treatment outcome, have not been thoroughly investigated [49]. Thus, in this review, we attempted to evaluate a number of aspects that influence how well the treatments work. Our primary objective was to use Sackett’s level of evidence, the Physiotherapy Evidence Database (PEDro) scale, and forest plots to evaluate the viability of the conclusions of all the included studies to better understand how EAATs affect the individual domains, such as the social, communicative, and behavioral abilities, as well as more the general functional outcomes, such as family functioning and quality of life. Our second objective was to assess whether the adjuvant therapy had a lasting effect after the horse intervention ended.

## 2. Methods

### 2.1. Search Protocol and Procedure

#### 2.1.1. Eligibility Criteria

To build upon the detailed, exact, and comprehensive reviews, our methodology for this systematic literature review followed the meta-analyses (PRISMA 2020) guidelines [50] and PROSPERO registration requirements (CRD42022363685). We screened the studies based on the following search terms and their variants and synonyms: autism spectrum disorder and equine-assisted activities and therapies (see Section A.1 for details). Moreover, to ensure the inclusion of all apt and precise articles that fulfilled the requirements, two authors simultaneously independently screened the publications based on the same eligibility criteria (see Table 1 for details). After screening all the included articles, a third author independently compared the studies screened by the first two authors. If the two authors had different opinions, then the three authors simultaneously reviewed the full texts of the articles to determine whether they met the inclusion criteria. Although we did not include systematic reviews as one of the criteria for this review, we still evaluated the relevant systematic studies so as not to exclude any relevant studies from this review.

In addition, the participants in the eligible studies must have had a diagnosis of ASD according to the Diagnostic and Statistical Manual of Mental Disorders (DSM) or have previously been diagnosed with autism in a qualified hospital or medical center. If the study included patients other than those with ASD (such as those with ADHD or ID), then it met the inclusion criteria; if no autistic patients were included, then the study was not eligible. Under the same criteria, in the included studies, we allowed trials without control groups; however, these trials needed to include the participation of live horses and not virtual or robot horses (simulated horseback riding), which we excluded.

#### 2.1.2. Information Sources

In the research domains, we searched all the articles from five electronic databases (PubMed, Scopus, ERIC, ProQuest, MEDLINE) that were related to psychology, health, education, physical training, clinical therapy, and neuroscience on 10 October 2022. We obtained the additional detailed supplementary information by utilizing the websites “connected papers” and “Google Scholar”. To find more specific articles, we also looked at the appropriate reference sections for any other relevant articles. Furthermore, we employed Review Manager 5 and Endnote X9 to archive and evaluate all the associated articles.

#### 2.1.3. Study Selection

In the database searches that corresponded to the search terms, we identified 365 publications between 2009 and 2022 (see Figure 1 for details). After we filtered the title and abstract screenings, 70 articles met the inclusion criteria, and 295 articles did not. In the screening process, we included 11 additional relevant articles from the references. Then, we selected a total of 81 articles to meet the inclusion criteria. Of the 81 articles that were initially eligible, 39 were duplicates that we needed to eliminate, while we reviewed the remaining 42 in their entireties. A total of 17 articles did not meet the criteria, and so we excluded them from the analysis. We included the other 25 articles. Notably, 6 articles provided sufficient and reasonable raw data for this meta-analysis, while we used the remaining 19 articles for the qualitative synthesis, as they did not provide raw data or lacked sufficient data (Section A.2, Table A1).

## 3. Data Extraction and Evaluation

### 3.1. Data Items

Two members of the research team used Excel-2212 (Microsoft, Redmond, Washington State, United States of America) and Review Manager 5.4.1 software (Cochrane, London, United Kingdom) to collect the following data: article data (first author, publication year, journal name, and country); participant data (sample demographics: age, gender, geographical information); sample features (diagnosis severity, diagnostic measure, other therapies, horseback riding experience); interventional and controlled characteristic data (handler accreditation, duration and session of program, terminology, comparison condition); outcome data (scales and subscales, scale validity, statistics including mean, standard deviation (SD), *p*-value, effect size) (see Table 2 and Table 3 for details).

### 3.2. Risk of Bias for Evaluation

We evaluated the internal validity and applicability of the RCTs and controlled clinical trials (CCTs) for the systematic review using the PEDro scale [51,52], which has 11 items. As usual, we considered PEDro scores from 6 to 10 to be high quality, scores from 4 to 5 to be fair quality, and scores of equal to or less than 3 to be low quality (see Table 4 for details). In addition, we used Sackett’s level of evidence [49], which can be used to sort individual studies into five levels of evidence, from Level I to Level V, for more accurate and reliable evaluations. RCTs are the best kind of evidence (Level 1), indicating clear relationships or conclusions. The lowest degree of evidence (Level 5) contains some single-case reports, narrative statements, or studies that do not indicate clear relationships or results (see Table 5 for details). We also looked at the “Risk of Bias Table” from Cochrane to evaluate the bias risk in these studies by looking up the criteria for each study and giving them a rating of low, high, or unclear in terms of risk (see Figure 2 for details).

**Table 2 ijerph-20-02630-t002:** Study characteristics.

Basic Article Information			Sample Characteristics								
			Sample Demographics						Diagnoses and Measures		
First Author, Year, Journal	Type of Trial	Reported Inclusion/ Exclusion of Participants (Y/N)	Final Sample of Participants (E, C)	Age (Range, Mean, SD)	Gender (Male/Female)	IQ (SD)	Participant Eligibility Criteria	Past/Recent EAI Experience	Screening Instrument for Participants	Diagnoses	Diagnostic Measure
Peters, B. C., 2022, [43], J Autism Dev Disord	RCTs	Y	*n* = 21 E(OTEE): 12 C: 9	6–13 THR: (-), 8.68 ± 2.09 RA: (-), 9.45 ± 1.62	Male: 16 Female: 5 THR: 10/2 CT: 6/3	NVIQ ≥ 55	(1) Aged 6–13 years old; (2) SCQ score ≥ 15; (3) Meets clinical cutoffs for ASD on ADOS, ADOS-2, or SRS-2; (4) Leiter-3 score ≥ 55; (5) Combined irritability and hyperactivity score ≥ 25	No EAA experience during previous 6 months.	SCQ ADOS-2 Leiter-3	ASD	HCC ABC-C SRS-2 GAS PEDI-CAT ASD
Zoccante, L., 2021, [25], J Clin Med	Pre–post design	Y	*n* = 15 Level 1: 7 Level 2: 6 Level 3: 2	7–15, 9.8 ± 2.2	Male: 13 Female: 2	(-)	ASD participants without: (1) Critical medical illness; (2) Previous experience with horses; (3) Distressed behavior.	N	ADOS-2 DSM-5	ASD ADHD: 9	VABS-2 DCDQ’07 PSI-SF IEMS
Zhao, M., 2021, [40], Int J Environ Res Public Health	RCTs	Y	*n* = 61 E(THR): 31 C(RA): 30	6–12; THR: (-), 7.06 ± 1.50 RA: (-), 7.13 ± 1.36	Male: 44 Female: 17 THR: 21/10 RA: 23/7	(-)	Children diagnosed with ASD, aged 6–12 years old.	(-)	DSM-5	ASD	SSIS-RS; ABLLS-R
Peters, B. C., 2020, [28], OTJR (Thorofare N J)	SCED	Y	*n* = 6	6–13	(-)	NVIQ ≥ 55 on Leiter-3	(1) Aged 6–13 years old; (2) Diagnosed with ASD on SCQ (≥15); (3) NVIQ ≥ 55 on Leiter-3; (4) Combined score ≥ 11 on irritability and hyperactivity subscales of ABC-C; (5) Meets physical, mental, and emotional standards set forth by PATH Intl.	No THR experience during previous 6 months	SCQ ADOS Leiter-3	ASD	VAS
Kalmbach, D., 2020, [53], Occup Ther Health Care	Explanatory sequential design	(-)	*n* = 4	8–13	Male: 4 Female: 0	NVIQ (M: 100, SD: 15)	(-)	(-)	ABAS	ASD	Semistructured interviews
Ozyurt, Gonca., 2020, [54], Montenegrin Journal of Sports Science and Medicine	RCTs	(-)	*n* = 24 E(EAA): 12 C: 12	4–12, 6.77 ± 1.3 EAA: 6.75 ± 0.7 C: 6.7 ± 0.64	Male: 17 Female: 7 EAA: 8/4 C: 9/3	(-)	(-)	No previous experience with equine-assisted activities.	(-)	ASD	CGAS FAD SCQ
Kwon, S., 2019, [55], Ann Rehabil Med	More group control	Y	*n* = 29 E(THR): 18 C(CT): 11	6–11 THR: 8.2 ± 1.7 CT: 7.5 ± 1.1	Male: 16 Female: 13 THR: 11/7 CT: 5/6	(-)	(1) Diagnosis of ASD or ID; (2) Aged 6–13 years old; (3) Body weight < 35 kg; height < 150 cm; (4) Understanding of simple instructions; (5) Appropriate physical development for rehabilitative horseback riding; (6) Informed consent from legal guardian.	N	(-)	ASD: 19 ID: 10	REVT PRES K-ABC-2 BSID-2 Luria Model
Pan, Z., 2018, [56], Front Vet Sci	RCTs	Y	*n* = 16 E(THR): 8 C(BA): 8	6–16 THR: 11.88 ± 2.45 BA: 9.80 ± 2.82	Male: 13 Female: 3 THR: 6/2 BA: 7/1	THR: 102.88 ± 16.28 BA: 100.25 ± 29.26	(1) Aged 6–16 years; diagnosis of ASD; (2) Combined total score of > 11 on irritability and stereotype subscales of ABC-C; (3) NVIQ score of ≥ 40 by Leiter-3.	(-)	SCQ ADOS-2 ABC-C Leiter-3	ASD CPD: 12 PM: 9 PD: 1 MD: 3 AD: 8 ADHD: 7 LD: 1	SALT SRS ABC-C Saliva cortisol
Gabriels, R. L., 2018, [57], Front Vet Sci	RCTs	Y	*n* = 64 E(THR): 36 C(BA): 28	6–16	(-)	NVIQ 85 or > 85		6 months	(-)	ASD	ABC-C SRS SALT
Tan, V. X., 2018, [58], J Autism Dev Disord	Case design	(-)	*n* = 6	3–14	Male: 1 Female: 5	NVIQ: 40 and 56	(-)	8 months–5 years	(-)	ASD	IPA
Harris, A., 2017, [59], Int J Environ Res Public Health	More group control	Y	*n* = 26 E(HR): 12 C: 14	6.08–9.33, 7.5 ± 10.57 HR: 8.2 ± 10.56 C: 7 ± 3.95	Male: 22 Female: 4	(-)	Excluded: (1) Not wearing helmet; (2) Known history of treating animals; (3) Fear or dislike of animals.	Four children in intervention group had more than 2–3 years.	Test in Social Communication Clinic	ASD	CARS2 ABC-C MOPI
Anderson, S., 2016, [60], J Autism Dev Disord	Case–control	Y	*n* = 15	5–16, 10 [3.8]	Male: 11 Female: 4	(-)	(1) Diagnosis of ASD; (2) No previous experience with horses.	N	DSM	ASD (27%) ADHD (20%) HSID (53%)	VABS ASQ EQ/SQ
Borgi, M., 2016, [61], J Autism Dev Disord	RCTs	Y	*n* = 28: E(EAT): 15 C: 13	6–12, 8.6 ± 1.7 EAT: 9.2 ± 1.8 CG: 8.0 ± 1.5	(-)	IQ > 70 EAT: 98.3 ± 16.2 CG: 92.8 ± 19.9	(1) Diagnosis of ASD, aged 6–12 years; (2) IQ > 70 on WISC-III; (3) Lack of previous THR experience.	Lack of previous therapeutic riding experience.	(-)	ASD	VABS TOL
Gabriels, R. L., 2015, [30], J Am Acad Child Adolesc Psychiatry	RCTs	Y	*n* = 116 E(THR): 58 C(BA): 58	6–16; THR: (-), 10.5 ± 3.2 BA: (-), 10.0 ± 2.7	Male: 101 Female: 15 THR: 49/9 BA: 52/6	NVIQ: THR: 86.7 [25.5] BA: 86.1 [22.7]	Leiter-R nonverbal IQ cutoff ≥ 40; SCQ-ASD screening cutoff ≥ 15; ABC-C score ≥ 11.	No more than two hours of EAAT within past six months.	SCQ ADOS-2 ABC-C Leiter-3	ASD MD AD ADHD, LD SD	PPVT-4 SALT SRS BOT-2 SIPT VABS ABC-C
Steiner, H., 2015, [62], Acta Physiol Hung	Control design	(-)	*n* = 26 E(THR): 13 C(PT): 13	10–13	Male: 12 Female: 14 THR: 6/7 PT: 6/7	(-)	(-)	(-)	(-)	ASD	APAS PAC-test
Holm, M. B., 2014, [63], J Autism Dev Disord	Single-case–control	Y	*n* = 3	6–8	Male: 3 Female: 0	(-)	(1) Diagnosis of ASD; (2) Aged 5–13 years old; (3) Available to participant in intervention; (4) Parental agreement.	Approximately one year.	(-)	ASD	KTEA-2 ABC-C CARS SRS SP-CQ
Lanning, B. A., 2014, [42], J Autism Dev Disord	Control design	Y	*n* = 18 * E(EAA): 10 C(SC): 8	4–15 EAA: 4–15, 7.5 ± 3.2 C: 5–14, 9.8 ± 3.2	Male: 21 Female: 4 EAA: 9/4 C: 12	(-)	(1) Diagnosis of ASD from physician or therapist; (2) Parental agreement; (3) No participation in EAA 6 months prior to start of study.	No EAA experiences during previous 6 months.	(-)	ASD	PedsQL CHQ
Ward, S. C., 2013, [38], J Autism Dev Disord	Quasi-experimental	Y	*n* = 21	8.1	Male: 15 Female: 6	(-)	(1) Meeting criteria for autism according to DSM IV-TR; (2) Qualified for services in public school division.	Thirteen participants with no TR experience.	DSM IV-TR CAB-T	ASD	GARS-2 SPSC
Jenkins, Sarah R., 2013, [64], Research in Autism Spectrum Disorders	Multiple baseline SCED	Y	*n* = 7 THR: 4 C: 3	6–14, 9.5	Male: 6 Female: 1	(-)	(1) No prior exposure to THR or hippotherapy; (2) Residing within 30 miles of primary research site.	No THR or HIP experience.	VABS-2	ASD TC Verbal and Motor Apraxia	CBC BOT-2
Ghorban, Hemati, 2013, [65], Journal of education and learning	Pre–post design	Y	*n* = 6	6–12	Male: 1 Female: 5	(-)	Meeting criteria for DSM-IV-TR.	(-)	(-)	ASD	TSSA
Gabriels, Robin L., 2012, [66], Research in Autism Spectrum Disorders	Waitlist control and pre–post design	Y	*n* = 42 THR: 26 C: 16	6–16, 8.7 E: 5–16, 8.6 C: 6–14, 8.8	Male: 36 Female: 6 E: 21/5 C: 15/1	NVIQ range of 44–139, Mean: 95.2	(1) Chronological ages of 6–16 years; (2) Diagnosis of autistic or Asperger’s disorder; (3) Combined score of at least 11 on ABC-C.	No THR experience within past three years.	ABC-C SCQ ADOS Leiter-R	ASD Asperger’s disorder Seizures	ABC-C BOT-2 SIPT VABS-2
Tabares, C., 2012, [67], Neurochemical Journal	Pre–post design	(-)	*n* = 8	8–16	Male: 8	(-)	(-)	(-)	(-)	ASD	ELISA
Janet K Kern., 2011, [68], Alternative Therapies in Health and Medicine	Pre–post design	Y	*n* = 24	3–12, 7.8 ± 2.9	Male: 18 Female: 6	(-)	(1) Between 3 and 12 years of age; (2) Primary diagnosis of ASD; (3) CARS score ≥ 30; (4) No previous participation in equine-assisted activities.	No previous EAA participation.	CARS	ASD	CARS Timberlawn Parent–Child Interaction Scales SP QLES-Q TSS
Bass, M. M., 2009, [69], J Autism Dev Disord	Case–control	Y	*n* = 34 E: 19 C: 15	5–10 E: 6.95 ± 1.67 C: 7.73 ± 1.65	Male: 29 Female: 5 E: 17/2 C: 12/3	(-)	(1) Meeting criteria for DSM-IV-TR.	No EAA experience.	DSM-IV-TR SSSS	ASD Asperger’s	SRS SP
Taylor, Renee R., 2009, [70], Occupational Therapy in Mental Health	SCED	(-)	*n* = 3	4–6	(-)	(-)	(1) Aged 4–6 years; (2) No other medical or psychiatric diagnoses.	(-)	(-)	ASD	PVQ

Table chronologically sorted and abbreviations alphabetically ordered as follows: (-): not reported; Y: yes; N: no; E: experimental group; C: control group; n/a: not applicable; n: number; BA: barn activity; Type of Trial: SCED: single-case experimental design; Final sample of participants: *n* = 18 *, initial sample = 25; BA: barn activities; CT: conventional therapy; EAAs: equine-assisted activities; EAT: equine-assisted therapy; HR: horseback riding; OTEE: occupational therapy in equine environment; PT: physical therapy; RA: regular activity; SC: social circle; THR: therapeutic horseback riding; Participant eligibility criteria: ABC-C: Aberrant Behavior Checklist—Community; NVIQ: nonverbal IQ; PATH Intl.: Professional Association of Therapeutic Horsemanship International; WISC-Ⅲ: Wechsler Intelligence Scale for Children III; Screening Instruments: ABAS: Adaptive Behavior Assessment System; ABC-C: Aberrant Behavior Checklist—Community; ADOS: Autism Diagnostic Observation Schedule; CAB-T: Clinical Assessment Battery Teacher Rating Form; CARS: Childhood Autism Rating Scale; DSM: Diagnostic and Statistical Manual of Mental Disorders; Leiter: Leiter International Performance Scale; SCQ: Social Communication Questionnaire; SSSS: Stone’s Social Skills Scale; VABS: Vineland Adaptive Behavior Scale; Diagnosis: AD: anxiety disorder; ADHD: attention-deficit/hyperactivity disorder; CPD: community psychiatric diagnoses; HSID: hypersensitivity and sensory integration disorder; ID: intellectual disability; LD: learning disability; MD: mood disorder; PD: psychotic disorder; SD: seizure disorder; TC: tuberous sclerosis; Diagnostic Measure: ABLLS-R: Assessment of Basic Language and Learning Skills—Revised; APAS: Ariel Performance Analysis System; BOT: Bruininks–Oseretsky Test of Motor Proficiency; BSID-2: Cognitive Domain of Bayley Scales of Infant Development 2; CBC: Child Behavior Checklist; CGAS: Children’s Global Assessment Scale; CHQ: Child Health Questionnaire; DCDQ’07: Developmental Coordination Disorder Questionnaire, as revised in 2007; ELISA: competitive enzyme immune essay method; EQ/SQ: empathizing quotient/systemizing quotient; FAD: McMaster Family Assessment Device; GARS-2: Gilliam Autism Rating Scale 2; GAS: Goal Attainment Scale; HCC: hair cortisol content; IEMS: interaction emotions motor skills; IPA: interpretive phenomenological analysis; K-ABC-2: Kaufman Assessment Battery for Children 2; KTEA-2: Kaufman Test of Educational Achievement—Second Edition; PAC-test: Pedagogical Analysis and Curriculum; PEDI-CAT ASD: Pediatric Evaluation of Disability Inventory Computer Adaptive Test, Autism Spectrum Disorder Module; MOPI: observational measure of child’s engagement; PedsQL: Pediatric Quality of Life 4.0 Generic Core Scales; PPVT-4: Peabody Picture Vocabulary Test, Fourth Edition; PRES: Preschool Receptive–Expressive Language Scale; PSI-SF: Parenting Stress Index—Short Form; PVQ: Pediatric Volitional Questionnaire; QLES-Q: Quality of Life Enjoyment and Satisfaction Questionnaire; REVT: Receptive and Expressive Vocabulary Test; SALT: Systematic Analysis of Language Transcripts; SIPT: Sensory Integration and Praxis Test; SP: Sensory Profile; SPSC: Sensory Profile School Companion; SRS: Social Responsiveness Scale; SSIS-RS: Social Skills Improvement System Rating Scale; TSIF: Test of Sensory Integration Function; TOL: Tower of London; TSS: Treatment Satisfaction Survey; TSSA: Triad Social Skills Assessment; VAS: Visual Analog Scale.

**Table 3 ijerph-20-02630-t003:** Intervention characteristics.

First Author, Year	Intervention Time				Intervention Information								
	Duration (Weeks)	Session Frequency (Per Week)	Session Time (min)	Total Therapy Time (Minutes)	Test at Pre-, Interim, and Postintervention	Terminology	Intervention Format (I or G)	Intervention Provider Accreditation	Other Therapy	Setting	Clients/Caregivers/ Animals	Interventional Components	Control Group Condition
Peters, B. C., 2022, [43]	10	(-)	60	(-)	At baseline and after intervention.	OTEE	G	AOTA, AHA, PATH	(-)	Riding center	Five occupational therapists, instructors, volunteers, leaders, side walkers, and horses.	(1) Greetings; (2) Activities with horses; (3) Goodbyes and caregiver debriefing.	Waitlist/OTGE
Zoccante, L., 2021, [25]	20	1	45	900; 50% individual sessions; 50% couple sessions.	Before and after 20 individual sessions.	EAAT	I/G	ASD①	Amateur Sports	Horse valley	One veterinarian, one horse assistant, two healthcare professionals, three horses.	Grooming, activities on ground, activities on horse	n/a
Zhao, M., 2021, [40]	16	2	≈60	≈1920	One week prior to intervention; at 8th interim week; after 16-week postintervention.	THR	G	IETC①	(-)	Outdoor and indoor arenas	Instructors, volunteers, and horses.	(1) Warm-up activities; (2) Instruction in riding skills and horsemanship skills; (3) THR exercises and activities; (4) Cool-down and reward activities.	RA
Peters, B. C., 2020, [28]	10	(-)	45–60	(-)	(-)	OTEE/HIP	I	PATH/AHA	School SLP, School OT	Riding center	Two occupational therapists, volunteers, leaders, side walkers.	(1) Greetings; (2) Ground and mounted activities; (3) Parent debriefing and goodbyes.	n/a
Kalmbach, D., 2020, [53]	10	1	45–60	450–600	1: From four to six weeks after intervention; 2: Six months after intervention.	OTEE	I/G	AOTA	SLP, PT, OT	Riding arena	Three researchers, occupational therapists, side walkers, volunteers, horses.	(1) Premounting segment; (2) Mounted segment; (3) Postmounting segment.	(-)
Ozyurt, Gonca., 2020, [54]	8	1	60	480	Pretesting and post-testing.	EAA	G	PATH	RT, special education	Riding center	Clinicians, educators, occupational therapist, physical therapist, therapeutic riding instructor, speech and language therapists, pediatrician, horses.	(1) Preparation and warmup; (2) Grooming and feeding; (3) Mounting and dismounting; (4) Horsemanship activities; (5) Finishing.	RT
Kwon, S., 2019, [55]	8	1	30	240	Before and after intervention.	THR	G	(-)	CT	Riding center	Riders, instructors, national licenses, leaders, side walkers, and horses.	(1) Stretching exercises; (2) Riding skills and riding; (3) Interaction with horses (such as brushing, feeding, putting stickers on them).	CT
Pan, Z., 2018, [56]	10	(-)	45	(-)	SALT: within one month pre- and postintervention; SRS: within 1 month pre- and postintervention; ABC-C: after 10-week intervention; SCDC: before each THR session and 20 min following each session.	THR	G	PATH	(-)	Riding center	Research staff, trained volunteers, horses.	(1) Saliva collection; (2) Sitting with a volunteer; (3) Starting group; (4) Reviewing group schedules; (5) Warm-up exercises; (6) Lesson and activity; (7) Cool-down exercises; (8) THR group dismount and thanking horses, all groups, and volunteers; (9) Drawing activity at table (20 min); (10) Saliva collection.	BA
Gabriels, R. L., 2018, [57]	10	1	45	450	(1) Baseline assessment; (2) Postintervention assessment; (3) 6-month postintervention follow-up assessment.	THR	G	PATH	Same as in Gabriels R. L. (2015)	Riding center	THR: leader, side walker, instructors, and horses; BA: volunteers.	Same as in Gabriels R. L. (2015).	BA
Tan, V. X., 2018, [58]	at least 4	1	(-)	(-)	After intervention	EAI: 5 TR: 1	I	(-)	SOT physiotherapy ST SST	(-)	Mental health professional	(-)	n/a
Harris, A., 2017, [59]	First Class: 7 Second Class: 5	1	45	225–315	Before and after intervention over approximately 7 weeks; MOPI at end of each session.	THR	G	BHS	Speech and language therapy; CPI/T; OT.	Horse-riding facility	Instructors, side walkers, minorities, teaching staff, volunteers, and horses.	(1) Preparation and mounting; (2) Riding skills and exercises; (3) Stretching exercises; (4) Thanking instructors and horses.	Waitlist
Anderson, S., 2016, [60]	5	1	180	900	First and last days of EAA intervention.	THR	G/I	BHS RDA	N	Horse center	Instructors, volunteers, and horses.	(1) Health and safety briefing; (2) Parental self-assessments and interviews; (3) Horsemanship activities, including grooming, leading, and mucking out; (4) Therapeutic riding.	Horsemanship/ stable management
Borgi, M., 2016, [61]	6 months	1	60–70	1500–1750	30 days before EAT sessions and 6 months after intervention.	EAT	G	FISE	CT and SA	Riding center	Instructors, expert veterinarian, 20 horses.	(1) Grooming and hand-walking horses; (2) Horseback riding; (3) Closure, feeding and saying goodbye to horses and group.	Waitlist
Gabriels, R. L., 2015, [30]	10	1	45	450	ABC-C and SRS measures assessed 1 month pre- and postintervention.	THR	G	PATH	Psychotropic medicine	Riding center	Leaders, volunteers, instructors.	Warm up; therapeutic riding skills (mounting, halting, steering, running, trotting); horsemanship skills (how to lead and care for horse); cool down.	BA
Steiner, H., 2015, [62]	(-)	(-)	30	(-)	Before and after one month of therapy; after three-month break (without therapy).	THR	G	(-)	Pedagogical sessions	(-)	Leader, assistants, horses.	(1) Warm-up exercise of stretching on horseback while horses not moving; (2) Horseback riding.	PT
Holm, M. B., 2014, [63]	Phase A: 4 Phase B: (-) Phase A’: (-)	Phase A: 1 Phase B: 1, 3 or 5 Phase A’: 1	30–45	(-)	1-month postintervention.	THR	I	NARHA	Medical therapeutics	Riding center	Walkers, leader, instructors, horses.	(1) Grooming, emphasizing touch, naming of parts, and following instructor; (2) Riding session.	THR dose (1 time/week)
Lanning, B. A., 2014, [42]	12	1	≈60	≈720	3, 6, 9, and 12 weeks.	EAA	G	PATH	(-)	Riding center	Psychology student trainees, occupational therapist, physical therapist, pediatrician or family physician, instructors, side walkers, horses.	(1) Basic safety lessons; (2) Grooming lessons; (3) Riding activities.	SC
Ward, S. C., 2013, [38]	6-week TR 6-week break 4-week TR 6-week break 8-week TR	(-)	40–45	(-)	Prior to intervention and Weeks 6, 16, 23, 26, and 30.	TR	G	PATH	Speech services OT: 1 PT: 1	Cori Sikich Therapeutic Riding Center	Coordinator, instructor, leader, two side walkers, rider pai, and horses.	(1) Orientation; (2) Mounting and riding; (3) Riding skills; (4) Closure.	(-)
Jenkins, Sarah R., 2013, [64]	9	1	60	540	Before, weekly during THR, and after 9-week therapy program.	THR	G	PATH	(-)	Horse arena	Leader, side walker, instructor.	Creating lesson plans based on each rider’s skill level and acquisition of target horsemanship skills.	Waitlist
Ghorban, Hemati., 2013, [65]	4	2	45	360	Before and after intervention.	THR	G	(-)	(-)	Horseback riding	Trainers, parents, teachers.	(1) Familiarity stage; (2) Practices; (3) Riding skills; (4) End of riding stage.	n/a
Gabriels, Robin L., 2012, [66]	10	1	60	600	Within one month of THR and one month after THR.	THR	G	PATH	Psychoactive medications	Riding center	Clinical psychologist, instructor, occupational therapist, volunteers.	(1) Putting on riding helmets; (2) Sitting and waiting on bench; (3) Mounting horses; (4) THR activities; (5) Dismounting horses; (6) Grooming horses; (7) Putting away equipment.	Waitlist
Tabares, C., 2012, [67]	4	1	(-)	(-)	Before and after hippotherapy sessions.	HIP	I	AZE	(-)	Equestrian center	(-)	(1) Making contact with animals; (2) Mounting horses; (3) Exercise ring; (4) Dismounting horses; (5) Saying goodbye.	n/a
Janet K Kern., 2011, [68]	24	1	60	1440	(1) Before beginning 3–6-month waiting period; (2) Before starting riding treatment; (3) After 3 months; (4) Within 6 months of riding.	EAA	G	(-)	(-)	Riding center	Parents, caregivers, health professionals, instructor, horses.	Leading, grooming, and tacking responsibilities.	n/a
Bass, M. M., 2009, [69]	12	1	60	≈720	Pretesting and post-testing.	THR	G	(-)	CT: 11	Good Hope Equestrian Training Center	Instructors, side walkers, volunteers, horses.	(1) Mounting and dismounting; (2) Warm-up exercises to stretch bodies; (3) Riding skills; (4) Mounted games; (5) Horsemanship activities.	Waitlist
Taylor, Renee R., 2009, [70]	16	1	45	720	Before, during, and after hippotherapy.	HIP	I	(-)	None	Riding facility	Occupational therapists, pediatric physical therapist, leader, side walkers.	(1) Donning of helmets and mounting preparation; (2) Riding and dismounting.	n/a

Table chronologically sorted and abbreviations alphabetically ordered as follows: I: individuals; G: group; C: controlled; AC: activity control; SCDC: saliva collection and determination of cortisol; Terminology: EAATs: equine-assisted activities and therapies; EAAs: equine-assisted activities; EAI: equine-assisted intervention; EAT: equine-assisted therapy; HIP: hippotherapy (equine-assisted occupational or physical therapy); OTEE: occupational therapy in equine environment; THR: therapeutic horseback riding; TR: therapeutic riding; HR: horse riding; Other Therapy: CPI/T: continued previous intervention/therapy; CT: conventional therapy; OT: occupational therapy; RT: regular therapy; SA: scholastic assistance; SLP: speech–language pathology; SOT: speech and occupational therapy; SST: Samoans sound therapy; ST: speech therapy; Intervention Provider Accreditation: AHA: American Hippotherapy Association; AOTA: American Occupational Therapy Association; ASD①: Associazione Sportiva Dilettantistica; AZE: Association of Zootherapy of Extremadura; BHS: British Horse Society; IETC①: International Equestrian Training Center, China; IFES: Italian Federation of Equestrian Sports; NARHA: North American Riding for the Handicapped Association; PATH: Professional Association of Therapeutic Horsemanship International; RDA: Riding for Disabled; Control Group Condition: OTGE: occupational therapy garden environment; SC: social circle, providing educational and recreational activities.

**Table 4 ijerph-20-02630-t004:** PEDro scoring for RCTs.

First Author, Year	Specified Eligibility Criteria	Random Subject Allocation	Allocation Concealment	Baseline Similarity of Groups	Blinding of Subjects	Blinding of Therapies	Blinding of Assessors	Measures of Outcomes	Intention to Treat	Between-Group Comparisons	Point and Variability Measures	Total
Peters, B. C., 2022, [43]	Y	Y	(-)	Y	(-)	(-)	Y	Y	Y	Y	Y	7
Zhao, M., 2021, [40]	Y	Y	(-)	Y	(-)	(-)	N	Y	Y	Y	Y	6
Ozyurt, Gonca., 2020, [54]	N	Y	(-)	(-)	(-)	(-)	Y	Y	Y	Y	Y	5
Pan, Z., 2018, 55]	Y	Y	(-)	Y	(-)	(-)	Y	Y	Y	Y	Y	7
Gabriels, R. L., 2018, [57]	Y	Y	(-)	Y	(-)	N	Y	Y	Y	Y	Y	7
Borgi, M., 2016, [61]	Y	Y	(-)	Y	(-)	(-)	(-)	Y	Y	Y	Y	6
Gabriels, R. L., 2015, [30]	Y	Y	(-)	Y	(-)	N	N	Y	Y	Y	Y	6
Bass, M. M., 2009, [69]	Y	Y	(-)	Y	(-)	(-)	(-)	Y	Y	Y	Y	6

**Scores:** 6–10 (high quality); 4–5 (fair quality); (-), not report; Y: yes; N: no.

**Table 5 ijerph-20-02630-t005:** Trial quality.

First Author, Year	Study Methods	C-Group Allocation	Report: Past/ Recent Riding Experience	Report: Screening of Riding Center	Report: Treatment Fidelity/ Integrity	PEDro Score	Level of Evidence
Peters, B. C., 2022, [43]	RCTs	Randomized	Y	Y	Y	7	I
Zoccante, L., 2021, [25]	Pre–post design	n/a	Y	Y	(-)	n/a	III
Zhao, M., 2021, [40]	RCTs	Randomized	(-)	Y	(-)	6	I
Peters, B. C., 2020, [28]	SCED	n/a	Y	Y	Y	n/a	IV
Kalmbach, D., 2020, [53]	Explanatory sequential design	n/a	(-)	Y	Y	n/a	IV
Ozyurt, Gonca., 2020, [54]	RCTs	Randomized	Y	Y	Y	5	II
Kwon, S., 2019, [55]	More group control	Nonrandomized but testing of baseline similarity	N	Y	Y	n/a	II
Pan, Z., 2018, [56]	RCTs	Randomized	(-)	Y	Y	7	I
Gabriels, R. L., 2018, [57]	RCTs	Randomized	(-)	Y	Y	7	I
Tan, V. X., 2018, [58]	Case design	n/a	Y	(-)	(-)	n/a	V
Harris, A., 2017, [59]	More group control	Nonrandomized but testing of baseline similarity	Y	Y	Y	n/a	II
Anderson, S., 2016, [60]	Pre–post design	Nonrandomized but testing of baseline similarity	Y	Y	Y	n/a	III
Borgi, M., 2016, [61]	RCTs	Randomized	Y	Y	Y	6	I
Gabriels, R. L., 2015, [30]	RCTs	Randomized	(-)	Y	Y	7	I
Steiner, H., 2015, [62]	Control design	Randomized	(-)	(-)	(-)	n/a	III
Holm, M. B., 2014, [63]	SCED	n/a	Y	Y	Y	n/a	IV
Lanning, B. A., 2014, [42]	Control design	Randomized	Y	Y	(-)	n/a	III
Ward, S. C., 2013, [38]	Quasi experimental interrupted-time-series design	Based on public classroom assignment and testing of baseline similarity	Y	Y	Y	n/a	III
Jenkins, Sarah R., 2013, [64]	SCED	Nonrandomized but testing of baseline similarity	N	(-)	Y	n/a	IV
Ghorban, Hemati., 2013, [65]	Pre–post design	n/a	(-)	(-)	(-)	n/a	IV
Gabriels, Robin L., 2012, [66]	Waitlist control and pre–post design	Testing of baseline similarity	Y	Y	Y	n/a	II
Tabares, C., 2012, [67]	Pre–post design	n/a	(-)	Y	Y	n/a	IV
Janet K Kern., 2011, [68]	Pre–post design	n/a	Y	(-)	Y	n/a	IV
Bass, M. M., 2009, [69]	RCTs	Randomized	Y	Y	(-)	6	I
Taylor, Renee R., 2009, [70]	SCED	n/a	(-)	(-)	Y	n/a	IV

## 4. Results

### 4.1. Description of Study

After careful screening and analysis, we gathered 25 eligible articles between 2009 and 2022: 2009 (*n* = 2); 2021 (*n* = 1); 2012 (*n* = 2); 2013 (*n* = 3); 2014 (*n* = 2); 2015 (*n* = 2); 2016 (*n* = 2); 2017 (*n* = 1); 2018 (*n* = 3); 2019 (*n* = 1); 2020 (*n* = 3); 2021 (*n* = 2); 2022 (*n* = 1). Researchers have conducted 25 studies in 10 countries, performing most of them in the United States (*n* = 14) [28,38,42,43,53,56,57,62,63,64,66,68,69,70]. They performed the others in the following countries: the United Kingdom (*n* = 2) [59,60]; Italy (*n* = 2) [25,61]; China (*n* = 1) [40]; Iran (*n* = 1) [65]; Australia (*n* = 1) [58]; Spain (*n* = 1) [67]; Hungary (*n* = 1) [62]; Turkey (*n* = 1) [54]; Korea (*n* = 1) [55].

### 4.2. Sample Characteristics

#### Individuals with ASD

Of all the included studies, the range of participants with ASD was 3–116, producing a total final sample size of 623, with an average sample size of 25. The largest sample included 115 participants [30], while the smallest sample included only three people [63,70]. The median sample size was 21, and 12 studies (48%) had sample intervals that focused on 15–29 participants. Meanwhile, the researchers reported the age of the demographic factors in 25 studies. The age range was 3–16 years old, and in four studies, the authors did not note the gender factor [25,47,56,60]. In almost all the studies (84%, *n* = 21), there were more male participants than female participants, which is consistent with the age and gender differences in the pathology of autism. In addition, in two trials, the ratios of male-to-female patients were approximately equal [55,62], and in two studies, the number of females was higher than the number of males [58,65]. Other than age and gender, researchers reported the nonverbal intelligence quotient in nine studies [25,27,37,49,52,53,56,60,69]. Additionally, in eight studies, the authors reported that all patients diagnosed with autism had also been diagnosed with other disorders or conditions, such as ADHD, ID, hypersensitivity and sensory integration disorder (HSID), learning disability (LD), seizure disorder (SD), etc. (see Table 2 for details).

### 4.3. Intervention Characteristics

#### 4.3.1. Screening Criteria and Instrument

In a total of 19 of 25 articles (76%), the authors reported the participant eligibility criteria; however, in six articles, they did not specify the criteria in detail for the population inclusion in the trials. Among the inclusion criteria, in 18 studies (72%), the authors specifically indicate whether the participants had previously participated in EAAT programs or had horseback riding experience. Although in no studies do the authors clearly point out whether the riding experience affected the experimental treatment effect, it is a key factor that can be explored to determine whether EAATs have lasting therapeutic effects, and we hope that researchers will test this hypothesis in future trials. 

The researchers most frequently used the Autism Diagnostic Observation Schedule (ADOS) (*n* = 6), Social Communication Questionnaire (SCQ) (*n* = 5), Leiter International Performance Scale (Leiter) (*n* = 5), and DSM (*n* = 5) during the participant screening phase. In the remaining studies, the authors used screening scales such as the Aberrant Behavior Checklist—Community (ABC-C), Adaptive Behavior Assessment System (ABAS), Clinical Assessment Battery Teacher Rating Form (CAB-T), Vineland Adaptive Behavior Scale (VABS), Childhood Autism Rating Scale (CARS), and Stone’s Social Skills Scale (SSSS). (See Table 2 for details).

#### 4.3.2. Intervention Dose 

In addition to the abovementioned factors, the trial session, frequency, and total time were also paramount. Of the twenty-five included studies, eight studies lacked data on the total length of the trials, while the authors reported the durations in the remaining 17 (68%) trials. The total durations of the programs ranged from 240 to 1920 min. The average length of the trials was approximately 756 min, with a median time of 600 min, and 11 studies had trials from 450 to 900 min. Whereas the shortest trial was 4 weeks and the longest was 25 weeks, with a mean of 11 weeks and median of 10 weeks, seven of the twenty-two trials (32%) in which the authors reported the program weeks had set-up times of 10 weeks (see Table 3 for details).

#### 4.3.3. Terminology

In these studies, the authors use different intervention terminologies, and the most used is THR (*n* = 13, 52%), while the remaining authors use terms such as occupational therapy in an equine environment (OTEE) (*n* = 3), EAA (*n* = 3), HIP (*n* = 2), EAT (*n* = 1), EAI (*n* = 1), TR (*n* = 1), and EAAT (*n* = 1). Although the names of the terminologies are different, the treatments that they describe primarily include the same training stages: warming up (health and safety briefings, stretching exercises, and so on), riding and horsemanship skills (mounting, halting, steering, running, trotting, brushing, feeding, and putting stickers on their horses), and cool-down and reward activities (thanking instructors and horses) (see Table 3 for details).

### 4.4. Study Methods and Trial Quality

Eight of the included articles were RCTs, and all but one of them (of fair quality) were of high quality. Notwithstanding, we noticed that in all eight RCTs, the authors did not report the allocation concealment or participant blinding because it was difficult to conceal the trial groupings from the participants in such trials. Additionally, the authors of 10 of the 25 studies that were included in this review randomly allocated the control groups to the experimental groups, whereas in the remaining nine studies, the authors did not have control groups and generally used within-group pre and post designs. A total of five of these nine trials were designed for individuals as opposed to groups. A total of 11 studies (44%) scored at Level II or higher when we assessed all 25 studies using Sackett’s level of evidence, which further indicated the high dependability of the studies used in this review (see Table 4 and Table 5 for details). To thoroughly analyze the trustworthiness of this systematic review, we also utilized the Cochrane risk-of-bias tool to evaluate three different types of risk (low risk, unclear risk, and high risk) for the six subdomains: (1) random sequence generation; (2) allocation concealment; (3) blinding of participants and personnel; (4) blinding of outcome assessment; (5) incomplete outcome data; (6) selective reporting. None of the 25 studies were low risk, and performance and detection biases were the most common (see Figure 2 for details), which is consistent with the nature of such trials, and which made it challenging to double-blind the participants and assessors, revealing an improvement direction for future trials, in which researchers should aim to blind the assessors. 

## 5. Outcome Measures and Effects

### 5.1. Outcome Measures

In only five of the twenty-five research studies do the authors report the intragroup effects. Authors report both the inter- and intragroup effects in a total of thirteen studies, and seven studies lack sufficient data to identify the effects. In numerous studies in this review, the authors evaluated the EAAT impact on multiple ability categories using a collection of subjective and objective measurements, including standardized tests graded by experts or parents, qualitative observational measures, physiological indicators, etc. Specifically, in 15 of the 25 studies, the authors employed caregiver-rated measures, such as the Social Responsiveness Scale (SRS), ABC-C, Sensory Profile (SP), Assessment of Basic Language and Learning Skills—Revised (ABLLS-R), Pediatric Volitional Questionnaire (PVQ), Triad Social Skills Assessment (TSSA), VABS, and ABAS. In six studies, the authors utilized standardized tests administered by trained clinicians or experts, such as the CARS, Sensory Integration and Praxis Test (SIPT), and Bruininks–Oseretsky Test of Motor Proficiency (BOT). In three studies, the authors used semi-structured interviews and observational measures to analyze the performance improvements during the training sessions. In three studies, the authors incorporated physiological parameters, such as the salivary hormone levels, and in one study, they evaluated the EAAT effects using a computer-assisted methodology (see Table 3, Table 6 and Table 7 for details).

### 5.2. Severity of ASD

The key criterion that we utilized to assess the effectiveness of the intervention was the improvement in the ASD severity. In this comprehensive review, in three studies that reported the ASD severity effects, the authors used the following four distinctive standardized measures: (1) the CARS; (2) the autism spectrum quotient (ASQ); (3) the Beck Depression Inventory (BDI); (4) the Children’s Global Assessment Scale (CGAS). They also used an additional qualitative observational scale: interpretive phenomenological analysis (IPA).

In a 2011 study [68] in which the authors used the CARS, they discovered that the EAAT program reduced the ASD severity in the participants. The findings are consistent with those of studies conducted in 2013 and 2017 in which the authors used the CARS [38,59], as well as with those of studies [54] in which the authors used the CGAS. Using IPA observed by parents, in a 2018 study, the authors illustrated the EAI impact on autism from a qualitative perspective, revealing that the EAI improved the broad aspects of overall psychosocial functioning [58] and for which the psychological and emotional satisfactions were consistent with those of the previous 2011 and 2014 studies [42,71] but slightly different from those of another previous study [64]. Overall, there is only limited evidence that EAAT programs are successful at reducing the autism severity in patients given the paucity of adequate primary data (see Table 6 for details).

**Table 6 ijerph-20-02630-t006:** Study results.

First Author, Year	Tested Domains/Variables	Type of Effect (B/W)	Type of Trial	Reported Measures/Variables Showing Substantial Improvement	Reported Measures/Variables Showing No Substantial Improvement	Results			
						Baseline Group (M, SD/SE)	Intervention Group (M, SD/SE)	Control Group (M, SD/SE)	Magnitude of Reported ES
Peters, B. C., 2022, [43]	Self-regulation, social communication, and social play.	B/W	Qualitative and quantitative research	Improved goal attainment and social motivation and decreased irritability.	Social communication, hyperactivity.	GAS: Primary goals: IG: 2.00, CG: 2.00; averages of goals: IG: 2.00, CG: 2.00 ABC: Irritability: IG: 14.65 (6.99), CG: 17.68 (4.69) SRS-2: Social motivation: 69.85 (9.39), CG: 74.67 (8.20)	GAS: Primary goal: 0.75 (1.45) Average of all goals: 0.39 (1.13) ABC: Irritability: 12.00 (5.89) SRS-2: Social motivation: 66.75 (12.39)	GAS: Primary goal: 0.00 (1.22) Average of all goals: −0.48 (1.03) ABC: Irritability: 15.53 (6.84) SRS-2: Social motivation: 71.00 (7.86)	GAS: Primary goal: *p* < 0.001 Average of all goals: *p* < 0.001 ABC: Irritability: *p* = 0.040 SRS-2: Social motivation: *p* = 0.033
Zoccante, L., 2021, [25]	Adaptive behavior, neuromotor function, and parent–child interaction.	W	Qualitative research	Greater adaptive behavior and coordination with increasing complexity of positive behavioral support.	Reduced parental distress.	Vineland-II: Communication: 48.1, SE [6.5] Socialization: 55.5, SE [4.9] Daily living skills: 60.5, SE [5.0] Motor skills: 66.9, SE [8.3] PSI-SF: Total score: 86.4, SE [4.3] Parental distress: 30.5, SE [2.4] PCDI: 25.7, SE [1.4] Difficult child: 29.6, SE [1.7] DCDQ’07: Total score: 37.5, SE [2.4]	Vineland-II: Communication: 57.5, SE [6.4] Socialization: 63.0, SE [5.4] Daily living skills: 72.5, SE [5.2] Motor skills: 83.6, SE [6.9] PSI-SF: Total score: 87.7, SE [6.0] Parental distress: 30.1, SE [2.6] PCDI: 26.2, SE [1.9] Difficult child: 32.4, SE [1.9] DCDQ’07: Total score: 40.2, SE [2.1]	n/a	Vineland-II: Communication: EAAT *p* < 0.001 Socialization: EAAT *p* < 0.001 Daily living skills: EAAT *p* = 0.01 Motor skills: EAAT *p* < 0.001 PSI-SF: Total score: *p* = 0.67 Parental distress: *p* = 0.69 PCDI: *p* = 0.62 Difficult child: *p* = 0.03 DCDQ’07: Total score: *p* = 0.01
Zhao, M., 2021, [40]	Social behavior change; communication skills.	B/W	Qualitative research	Overall social interaction, communication, responsibility, and self-control.	(-)	SISS: Social skills: IG: 44.68 ± 7.48, CG: 44.27 ± 4.31 Subdomains: Communication: IG: 6.71 ± 1.77, CG: 7.03 ± 1.54 Cooperation: IG: 7.55 ± 1.77, CG: 7.50 ± 1.41 Assertion: IG: 4.90 ± 1.58, CG: 4.63 ± 1.10 Responsibility: IG: 6.23 ± 1.23, CG: 5.87 ± 1.01 Empathy: IG: 5.42 ± 1.29, CG: 5.70 ± 1.02 Engagement: IG: 6.65 ± 1.45, CG: 6.47 ± 1.14 Self-control: IG: 7.23 ± 1.73, CG: 7.07 ± 1.53 ABLLS-R: Social interaction scores: IG: 24.03 ± 3.38, CG: 24.13 ± 3.59	SISS: Social skills: Interim: 48.26 ± 6.51, Post: 50.87 ± 6.47 Subdomains: Communication: Interim: 7.74 ± 1.55, Post: 8.48 ± 1.86 Cooperation: Interim: 7.97 ± 1.66, Post: 8.16 ± 1.73 Assertion: Interim: 5.23 ± 1.52, Post: 5.71 ± 1.47 Responsibility: Interim: 6.74 ± 1.21, Post: 7.00 ± 1.24 Empathy: Interim: 5.68 ± 1.19, Post: 5.90 ± 1.27 Engagement: Interim: 7.52 ± 1.36, Post: 7.68 ± 1.51 Self-control: Interim: 7.39 ± 1.75, Post: 7.94 ± 1.55 ABLLS-R: Social interaction scores: Interim: 27.74 ± 2.66, Post: 33.84 ± 4.00	SISS: Social skills: Interim: 45.13 ± 4.67, Post: 45.43 ± 5.08 Subdomains: Communication: Interim: 7.17 ± 1.53 Post: 7.27 ± 1.46 Cooperation: Interim: 7.57 ± 1.30 Post: 7.63 ± 1.22 Assertion: Interim: 4.80 ± 1.19, Post: 5.07 ± 1.39 Responsibility: Interim: 6.33 ± 1.21, Post: 6.13 ± 1.17 Empathy: Interim: 5.60 ± 1.10, Post: 5.53 ± 1.17 Engagement: Interim: 6.90 ± 1.09, Post: 7.03 ± 1.19 Self-control: Interim: 6.77 ± 1.55, Post: 6.77 ± 1.55 ABLLS-R: Social interaction scores: Interim: 25.60 ± 3.52 Post: 25.87 ± 3.05	SISS: Social skills: Compared with CG: *p* < 0.05 Compared with pretest: *p* < 0.01 ABLLS-R: Social interaction scores: Compared with CG: *p* < 0.01 Compared with pretest: *p* < 0.05 or *p* < 0.01
Peters, B. C., 2020, [28]	Performance goals, behavior, and social functioning.	n/a	Qualitative research	Four participants reported improvements in irritability and hyperactivity.	Two participants reported improvements in irritability and hyperactivity.	n/a	n/a	n/a	n/a
Kalmbach, D., 2020, [53]	Parental perspectives on child’s experience of occupational therapy (one) and its impact on child’s (two) and family’s daily lives (three).	n/a	Qualitative research	Occupational performance, social motivation, social communication, and self-regulation.	(-)	n/a	n/a	n/a	n/a
Ozyurt, Gonca., 2020, [54]	Social functioning, autistic behaviors, family functioning, and clinical severity.	B/W	Qualitative research	ASD severity and improvements in maternal mental health and family functioning.	Responsiveness and general functions.	CGAS: 57 ± 9.24 FAD: Communication: 2.5 ± 0.52 Role subscale: 2.31 ± 0.59 Involvement: 2.38 ± 0.58 Behavioral control: 2.23 ± 0.55 SCQ: 19.92 ± 4.12 BDI: 18.5 ± 6.31	CGAS: 61.83 ± 11.47 FAD: Communication: 2.2 ± 0.59 Role subscale: 1.88 ± 0.38 Involvement: 1.93 ± 0.59 Behavioral control: 1.93 ± 0.38 SCQ: 18.25 ± 3.70 BDI: 16.25 ± 5.46	FAD: Involvement: 2.42 ± 0.56 Behavioral control: 2.35 ± 0.47	CGAS: *p* = 0.0004 FAD: Communication: *p* = 0.001 Responsiveness: *p* > 0.05 Involvement: *p* = 0.01 Behavioral control: *p* = 0.01 General functions: *p* > 0.05 SCQ: *p* = 0.002 BDI: *p* = 0.0001
Kwon, S., 2019, [55]	Language function; cognitive function; intelligence and achievement.	W	Qualitative research	Significant improvements in receptive and expressive language and cognitive functions.	(-)	REVT Reception: IG: 17.44 ± 19.97, CG: 13.82 ± 18.81 BSID of Cognitive Domain: IG: 130.38 ± 21.87, CG: 136.00 ± 19.51	REVT Reception: 20.11 ± 20.84 BSID of Cognitive Domain: 133.69 ± 23.29	REVT Reception: 15.27 ± 18.12 BSID of Cognitive Domain: 138.33 ± 20.20	All domains statistically significant (*p* > 0.05)
Pan, Z., 2018, [56]	Adaptive skills and aberrant and social behaviors.	B/W	Qualitative research	Significant improvements in hyperactivity, social awareness, irritability, and communication behaviors.	Words or new words spoken.	ABC-C: Hyperactivity: IG 20.86 (12.13), CG 17.33 (4.46) SRS: Social awareness: IG 15.43 (3.95), CG 12.29 (2.56) Social communication: IG 41.00 (9.33), CG 29.29 (9.83)	ABC-C: Hyperactivity: 16.00 (8.64) SRS: Social awareness: 11.29 (1.38) Social communication: 34.57 (3.95)	ABC-C: Hyperactivity: 24.33 (6.02) SRS: Social awareness: 13.57 (4.12) Social communication: 31.29 (10.98)	ABC-C: Hyperactivity: *p* = 0.04 SRS: Social awareness: *p* = 0.01 Social communication: *p* = 0.03
Gabriels, R. L., 2018, [57]	Irritability, hyperactivity, social and communication behaviors.	B/W	Qualitative research	Reduction in irritability behavior Significant improvement in social and communication behaviors.	(-)	Irritability: IG 15.86 (9.52), CG 14.43 (8.69) Hyperactivity: IG 20.75 (20.71), CG 20.71 (20.75)	Irritability: 9.00 (8.08) Hyperactivity: 13.28 (17.07)	Irritability: 11.96 (9.29) Hyperactivity: 17.07 (13.28)	Irritability: *p* < 0.02; after 6 months: *p* = 0.52 Hyperactivity: *p* = 0.08; after 6 months: *p* = 0.2
Tan, V. X., 2018, [58]	Psychosocial outcomes.	n/a	Qualitative research	Self-concept, emotional wellbeing, self-regulatory ability, social benefits.	(-)	n/a	n/a	n/a	n/a
Harris, A., 2017, [59]	Symptomology and social functioning.	B/W	Qualitative research	Significant reduction in ASD symptom severity and hyperactivity.	Irritability, lethargy, stereotype, inappropriate speech.	CARS-2: IG 40.95 (6.07), CG 42.61 (7.52) ABC-C: Hyperactivity: IG 26.30 (10.73), CG 21 (1.07)	CARS-2: 40.05 (5.57) ABC-C: Hyperactivity: 22.30 (9.67)	CARS-2: 42.61 (7.52) ABC-C: Hyperactivity: 21 (11)	CARS-2: *p* = 0.013, ES = 0.5 ABC-C: Hyperactivity: *p* = 0.009, ES = 0.518
Anderson, S., 2016, [60]	Social functioning; behavior skills.	B/W	Qualitative research	Increasing empathy and reduction in maladaptive behaviors.	Communicative, socialization, systemizing quotient.				
Borgi, M., 2016, [58]	Adaptive and executive functioning.	B/W	Qualitative research	Social and executive functioning.	(-)				VABS: Socialization: *p* = 0.034 Motor skills: *p* = 0.021 TOL: Planning time: *p* = 0.026
Gabriels, R. L., 2015, [30]	Self-regulation; socialization; communication; adaptive, and motor behaviors.	B/W	Qualitative and quantitative research	Significant improvement in irritability and hyperactivity; social cognition and communication; total number of words and new words.	Lethargy/social withdrawal, stereotyping, inappropriate speech, social awareness, social motivation, autistic mannerism.	ABC: Irritability: IG 16.0 (9.84), CG 16.1 (9.80) Hyperactivity: IG 21.9 (10.7), CG 21.0 (9.69) SRS: Social cognition: IG 20.3 (5.63), CG 19.3 (5.58) Social communication IG 36.8 (10.04), CG 33.9 (8.84) SALT: Number of different words used: IG 104.6 (58.45), CG 119.1 (64.55) Number of words used: IG 219.2 (132.19), CG 277.6 (171.53)	ABC: Irritability: 9.5 (7.98) Hyperactivity: 14.3 (9.66) SRS: Social cognition: 17.6 (5.55) Social communication 30.2 (8.75) SALT: Number of different words used: 116.7 (66.00) Number of words used: 253.7 (154.62)	ABC: Irritability: 13.6 (10.08) Hyperactivity: 18.4 (10.26) SRS: Social cognition: 19.1 (5.64) Social communication 33.6 (11.38) SALT: Number of different words used: 118.4 (62.75) Number of words used: 270.5 (162.88)	Regulation of irritability: ES = 0.5, *p* = 0.002 Hyperactivity: ES = 0.53, *p* = 0.001 Social cognition: ES = 0.41, *p* = 0.05 Social communication: ES = 0.63, *p* = 0.003 Total number of words: ES = 0.54, *p* = 0.01 New words: ES = 0.54, *p* = 0.01
Steiner, H., 2015, [62]	Communication, self-care, motor skills, and socialization.	B/W	Qualitative and quantitative research	Communication, self-care, motor skills, and socialization.					
Holm, M. B., 2014, [63]	Parent-nominated target behaviors.	W	Qualitative research	Positive behaviors to be increased included eye contact, verbalization, and naming of people/items.					
Lanning, B. A., 2014, [42]	Social and emotional functioning.	B/W	Qualitative research	Physical, emotional, and social functioning.	Emotional and social functioning.				
Ward, S. C., 2013, [38]	Social communication and sensory processing skills.	W	Qualitative and quantitative research	Social interaction, sensory processing, and decreased severity of symptoms associated with ASD.	(-)				
Jenkins, Sarah R., 2013, [64]	Behavior	n/a	Qualitative and quantitative research	N	No clinically significant effects on mood, off-task behavior, problem behavior, compliance, or language.	n/a	n/a	n/a	n/a
Ghorban, Hemati., 2013, [65]	Social skills	W	Quantitative research	Initiating interactions and substantially maintaining interactions.	Responding interaction	n/a	n/a	n/a	Total score of social skills: sig. = 0.04 Subtest: Affective understanding/perspective taking and initiating interaction: sig. = 0.01 Maintaining interaction: sig. = 0.003
Gabriels, Robin L., 2012, [66]	Self-regulation, adaptive living skills, motor skills.	B/W	Quantitative research	Self-regulation behaviors	Receptive language	ABC-C: Irritability 20.2 ± 8.9 Lethargy 12.4 ± 7.7 Stereotypy 6 ± 4.2 Hyperactivity 23.7 ± 9.9 Adaptive skills: Raw social score 104.9 ± 29.9 Raw communication score 143.6 ± 24.9 Raw daily score 110.6 ± 35.1 Adaptive total score 75.5 ± 10.4 Motor skills BOT: 2 45.5 ± 15.5 SIPT: verbal score: 16 ± 7.2 SIPT: postural score: 19.5 ± 7.4	ABC-C: Irritability: 12.9 ± 8.5 Lethargy: 6.3 ± 7.1 Stereotypy: 3.3 ± 3.5 Hyperactivity: 17.1 ± 11.6 Adaptive skills: Raw social score: 113.2 ± 27.4 Raw communication score: 149 ± 24.8 Raw daily score: 117.4 ± 32.6 Adaptive total score: 79.2 ± 11.3 Motor skills BOT 2: 53.4 ± 15.2 SIPT: verbal score: 18.8 ± 7 SIPT: postural score: 22.9 ± 7.1		ABC-C: Irritability, lethargy, stereotypy, hyperactivity: *p* < 0.001 Adaptive skills: Raw social score: *p* = 0.016 Raw communication score: *p* = 0.035 Raw daily score: *p* = 0.011 Adaptive total score: *p* < 0.001 Motor skills BOT 2: *p* < 0.001 SIPT: verbal score: *p* < 0.001 SIPT: postural score: *p* = 0.009
Tabares, C., 2012, [67]	Hormonal changes	n/a	Quantitative research	Decreased salivary cortisol levels and increased progesterone.	(-)	Hormone cortisol: pre-hippotherapy: 2.79 ± 0.52 ng/mL Hormone progesterone: pre-hippotherapy: 28.63 ± 12.81 pg/mL	Hormone cortisol: post-hippotherapy: 4.015 ± 1.59 ng/mL The rest of the post-hippotherapy sessions: 2.23 ± 0.75 ng/mL Hormone progesterone: post-hippotherapy: 51.59 ± 33.11 pg/mL The rest of the sessions: 26.03 ± 11.98 pg/mL	n/a	All domains statistically significant (*p* ≤ 0.05)
Janet K Kern., 2011, [68]	Severity of autism symptoms, parent–child interactions.	W	Quantitative research	Severity of autism symptoms.	Parent–child interactions.				
Bass, M. M., 2009, [69]	Social functioning	B/W	Qualitative and quantitative research	Social function, greater sensory seeking, sensory sensitivity, social motivation, less inattention, distractibility, and sedentary behaviors.	Social cognition and social awareness.				
Taylor, Renee R., 2009, [70]	Motivation	n/a	Qualitative research	Volition	(-)	n/a	n/a	n/a	n/a

Abbreviation: B: between-group effect; W: within-group effect; CI: confidence interval; IG: intervention group; CG: control group.

**Table 7 ijerph-20-02630-t007:** Assessor types.

First Author, Year	Type of Assessment	Information Sources	Blinding of Assessors	Raters/Informants/			
				Authors/ Research Staff	Parent/ Caregiver	Staff/Instructor at Horse Center	Independent Raters
Peters, B. C., 2022, [43]	Expert and parent questionnaires/semi structured interviews/physiological measures	Parents, occupational therapists, authors	Partly Blind		Y	OTs	OTs
Zoccante, L., 2021, [25]	Parent questionnaires	Caregiver, clinical psychologist	(-)				Clinical psychologist
Zhao, M., 2021, [40]	Parent/expert questionnaires	Teachers at training center and parents	Nonblind		Y	Y	
Peters, B. C., 2020, [28]	Parent/expert questionnaires/visual analog scale	Parents	Nonblind		Y		
Kalmbach, D., 2020, [53]	Semi structured interviews	Parents	Nonblind	Y	Y		
Ozyurt, Gonca., 2020, [54]	Clinical and parent questionnaires	Parents and educators	Blind		Y		Clinician/educator
Kwon, S., 2019, [55]	Expert questionnaires/Luria model, battery	Speech and occupational therapists	(-)				Speech and occupational therapists
Pan, Z., 2018, [55]	Expert and parent questionnaires/saliva sample test	Caregiver, study personnel, speech therapist	Partly Blind	Y	Y		Speech therapist
Gabriels, R. L., 2018, [57]	Parent/expert questionnaires	Caregiver, speech therapist	Partly Blind		Y		Speech therapist
Tan, V. X., 2018, [58]	Semi structured interviews	Parents	Nonblind		Y		
Harris, A., 2017, [59]	Expert questionnaires	School teaching staff	(-)				School teaching staff
Anderson, S., 2016, [60]	Parent-reported questionnaires and semi structured tests	Parents	(-)		Y		
Borgi, M., 2016, [61]	Interviews	Parents	Blind		Y		
Gabriels, R. L., 2015, [30]	Parent/expert questionnaires	Study personnel, speech therapist, caregiver	Blinding to study personnel Unblinded caregiver questionnaires	Y	Y		
Steiner, H., 2015, [62]	APAS equipment/special test	Authors	Nonblind	Y			
Holm, M. B., 2014, [63]	Parent evaluation/ caregiver questionnaire/ videotaped and observed measures	Parents/caregiver/ leader	Nonblind	Y	Y	Y	
Lanning, B. A., 2014, [42]	Parent questionnaires	Parents/children	Nonblind		Y		
Ward, S. C., 2013, [38]	Parent rating/questionnaires	School group teacher and coordinator	Blind				School teacher
Jenkins, Sarah R., 2013, [64]	Observational assessment/parent questionnaires	Parents/observers/teachers	(-)		Y		
Ghorban, Hemati., 2013, [65]	Parent questionnaires	Parent or teacher	(-)		Y		
Gabriels, Robin L., 2012, [66]	Parent questionnaires	Parents/legal guardians, graduate student research assistants, occupational therapists	Nonblind	Y	Y	Y	
Tabares, C., 2012, [67]	Laboratory methods	Research staff	Nonblind	Y			
Janet K Kern., 2011, [68]	Parent- and clinician-rated measures	Research assistant/parents	Blind		Y		Research assistant
Bass, M. M., 2009, [69]	Social and communication skills	Parents or teachers	Blind		Y		School teacher
Taylor, Renee R., 2009, [70]	Observational assessment tool	OTgs	Blind			Y	OTgs

Abbreviations: (-): not reported; Y: yes; N: no; OTs: occupational therapies; OTgs: occupational therapy graduate students; APAS: Ariel Performance Analysis System.

### 5.3. Social Functioning

The social communication domain was the most crucial aspect of the generally reported autism impairment, with a total of 13 studies (52%) in which the authors describe the social and communication abilities following EAAT programs, using seven different standardized measures rated by parents or caregivers. In all the reported testing in the social domains, in five studies, the authors found considerable improvements in the overall SRS scores; however, the findings differed among the SRS subscales. More specifically, we analyzed the effects of EAAT programs on the social communication symptoms connected to ASD from four studies in which the authors provide exact data using the SRS [27,37,50,53]. According to the meta-analysis results, the EAATs more significantly improved the social functioning of the children with ASD than that of those in the control group (SMD = −0.33, 95% CI [−0.47, −0.19], *p* < 0.00001) (see Figure 3 for details). Among the five subdomains, social cognition (SMD = −0.47, 95% CI [−0.77, −0.18], *p* = 0.002) and social communication (SMD = −0.58, 95% CI [−0.91, −0.26], *p* = 0.0004) had significant improvements after the participants experienced EAAT programs; however, social awareness (SMD = −0.20, 95% CI [−0.49, 0.09], *p* = 0.17), social mannerisms (SMD = −0.14, 95% CI [−0.48, 0.21], *p* = 0.44), and social motivation (SMD = −0.26, 95% CI [−0.55, 0.03], *p* = 0.08) had no statistically significant differences. The findings of the meta-analysis are in line with those of earlier studies in which the authors found that the parents believed that their children were more driven to complete daily tasks and got along better with others after participating in the EAATs [47,63,69]. However, the outcomes of the subdomain improvements reported across the trials were not totally uniform in all the research. For instance, in a 2018 study with 14 samples [56], the authors found that EAAT substantially enhanced the social awareness domain in autistic children; however, based on the forest plot and according to the outcomes of the other three trials, it was not effective at enhancing the social awareness domain. In the studies on subdomains such as social cognition, the authors also report different results, which constitutes heterogeneity in the subdomains and overall.

The remaining six scales used to assess the social communication domain level include the SCQ, ABLLS-R, PVQ, CARS, TSSA, and Social Skills Improvement System (SSIS) rating scales. In a 2020 study [54] in which the authors utilized the SCQ, they demonstrated that the experimental group exhibited a considerable improvement in the social domain, while the control group had no substantial changes after 8 weeks of EAA. This was followed by a study in 2021 [40] in which the authors utilized the ABLLS-R and SSIS. They revealed substantial improvements in the social communication domain in the experimental group compared with the conventional group over 16 weeks. These findings, presented using multiple scales, were also largely compatible with the results displayed in the forest plot, which indicates that, in general, EAAT programs improve the social communication domains of individuals with ASD; however, the aspect and improvement degrees of the EAAT on the participants differed based on numerous parameters.

### 5.4. Language Ability

As another area of high impairment for individuals with autism, in four studies, the authors made use of four different standard assessments and reported improvements in the language skills of individuals with autism after trials. These four measures included the following: the Receptive and Expressive Vocabulary Test (REVT) (*n* = 1) [55]; the Preschool Receptive–Expressive Language Scale (PRES) (*n* = 1) [55]; the Systematic Analysis of Language Transcripts (SALT) (*n* = 3) [27,53,56]; the Peabody Picture Vocabulary Test, Fourth Edition (PPVT-4) (*n* = 1) [30].

Following the extraction of the raw data from a meta-analysis of two trials using the SALT, the experimental group’s language abilities had significantly improved compared with those of the comparison group following EAAT programs (SMD = 0.52, 95% CI [0.27, 0.77], *p* < 0.0001) (see Figure 4 for details). Although the 2018 trial [56] did not reveal significant gains in the two subdomains, the number of different words domain and number of words domain, according to another follow-up experiment performed in 2018 [57], there were substantial improvements in the two subdomains, and the authors also offer some evidence of the lasting effects of EAAT programs on enhancing the language abilities of people with autism, as they again measured the results with the SALT six months later and revealed non-substantial declines in the two subdomains. In another 2019 study in which the authors employed the REVT [55], they found considerable gains in the receptive linguistic knowledge. Overall, there is limited evidence to suggest that EAAT programs improve the language skills of individuals with ASD.

**Figure 3 ijerph-20-02630-f003:**
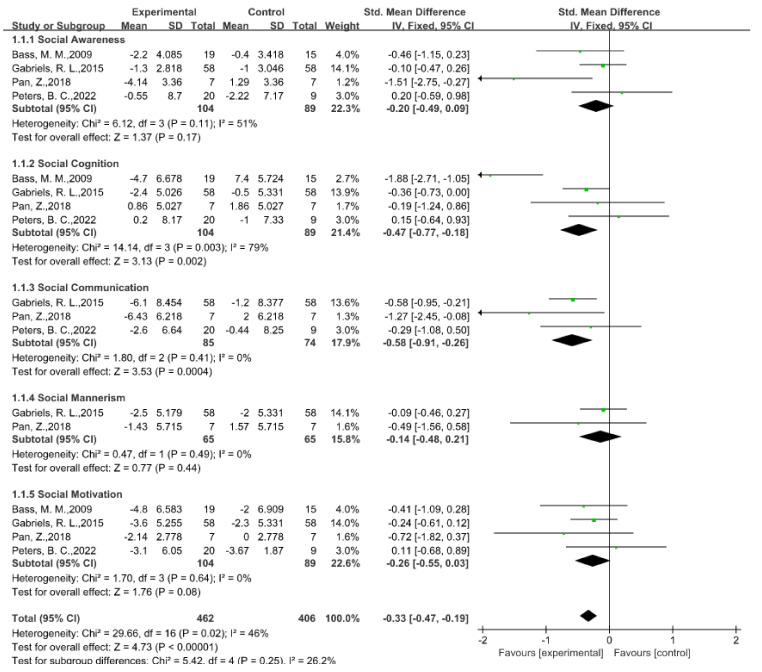
Forest plot of social function using SRS [30,43,56,69].

### 5.5. Behavioral Regulation

In a 2014 study, the authors found that self-regulation in children was linked to improvements in the executive function, sensory processing, and emotion regulation, all of which led to behavior problems [71]. Therefore, measuring and assessing the self-regulation and behavior of individuals with autism is one of the universal criteria for measuring ASD severity. In this review, we used seven different measurement tests to determine the behavioral regulation scores. The most common measurement standards for measuring the intervention effects on people with ASD are the ABC-C (*n* = 9) and VABS (*n* = 5). The five remaining scales and instruments that researchers used to measure behavioral competence are as follows: the ABAS-3 (*n* = 1) [28]; the Autism Spectrum Disorder Module (PEDI-CAT ASD) (*n* = 1) [43]; the Interaction Emotions and Motor Skills observational scale (IEMS) (*n* = 1) [25]; the Child Health Questionnaire (CHQ) (*n* = 1) [42]; the Child Behavior Checklist (CBC) (*n* = 1) [64]; physiological measures using salivary or hair cortisol (*n* = 3) [37,53,58]. These tools have high interverbal consistencies, good reliabilities, and well-established validities.

Specifically, five studies in which the authors used the ABC-C [27,37,53,56,59] provided us with sufficient and reliable data to conclude that the behavioral competence of individuals with ASD significantly improved after EAAT programs compared with the control group (SMD = −0.30, 95% CI [−0.42, −0.17], *p* < 0.00001) (see Figure 5 for details). Based on the subdomain analyses, the irritability (SMD = −0.44, 95% CI [−0.70, −0.19], *p* = 0.0007) and hyperactivity (SMD = −0.43, 95% CI [−0.69, −0.17], *p* = 0.001) had statistically significant improvements; however, there were no significant improvements in terms of lethargy (SMD = −0.21, 95% CI [−0.53, 0.11], *p* = 0.19), stereotypy (SMD = −0.04, 95% CI [−0.36, 0.27], *p* = 0.79), or inappropriate speech (SMD = −0.20, 95% CI [−0.52, 0.12], *p* = 0.21). 

The remaining four studies [27,49,61,62] in which the authors used the VABS lacked controlled trials; however, they do report pre- and post-trial data. We also used the fixed-effects model with 95% confidence intervals that the participants’ behavioral competency would substantially increase following the EAAT programs (SMD = 0.22, 95% CI [0.09, 0.35], *p* = 0.0008) (see Figure 6 for details). Participants also demonstrated appreciable improvements in the socialization subdomain following EAATs (SMD = 0.27, 95% CI [0.03, 0.52], *p* = 0.03). However, the data do not indicate any appreciable improvement for the other three domains: communication (SMD = 0.18, 95% CI [−0.06, 0.43], *p* = 0.15); daily living skills (SMD = 0.20, 95% CI [−0.06, 0.46], *p* = 0.13); adaptive behavior (SMD = 0.24, 95% CI [−0.06, 0.54], *p* = 0.11).

The results of the meta-analysis are basically consistent with the conclusions reported in previous studies. In five previous studies [25,36,49,52,55], the authors found that EAATs led to considerable improvements in self-control and decreases in negative behaviors in the experimental groups. However, in a 2013 [64] study in which the authors used direct observation to measure the effects, they found that a 9-week THR program did not substantially improve the mood, behavior, or communication abilities. Overall, there were enough data to conclude that EAAT programs enhance the behavioral abilities of ASD populations, despite the varied reports among the subfields.

### 5.6. Motor and Sensory Functions

In recent evidence-based investigations, the authors reveal that individuals with ASD exhibit motor and sensory deficits throughout their lifespans [49,72]. In nine studies, the authors implemented six different standard measures to assess the impacts of the EAAT programs on the motor and sensory functions of individuals with autism. Of these standard measures, researchers most frequently use the SP (*n* = 4) [25,50,51,55], BOT-2 (*n* = 2) [30,66], and SIPT (*n* = 2) [30,66]. The remaining scales include the CGAS (*n* = 1) [54], Developmental Coordination Disorder Questionnaire’07 (DCDQ’07) (*n* = 1) [25], and Sensory Profile School Companion (SPSC) (*n* = 1) [38]. In two studies [30,66], the authors employed the SIPT and BOT-2, which indicated changes in the motor and sensory functioning of children with ASD following their participation in EAATs; however, there was only one study in which the authors used a control group [30]. Thus, we compared the pre- and post-changes, revealing that the patients with autism who participated in EAAT programs appeared to have experienced significant improvements in their motor (SMD = 0.29, 95% CI [0.01, 0.58], *p* = 0.04) (see Figure 7 for details) and sensory (SMD = 0.29, 95% CI [0.10, 0.48], *p* = 0.003) abilities (see Figure 8 for details). Nevertheless, among the subcomponents, the verbal function did not significantly improve (SMD = 0.21, 95% CI [−0.07, 0.50], *p* = 0.14). Only the postural subdomain improved (SMD = 0.36, 95% CI [0.10, 0.62], *p* = 0.007). In three previous investigations [33,62,64], the authors confirmed the findings of the meta-analysis and demonstrated that the motor and sensory functioning domains substantially improved from the pre- to post-test in the experimental groups. However, according to another two studies [63,69], the EAAT programs had no clinical effects for ASD in this area. Overall, because only two articles provided enough data for the meta-analysis, there was only limited evidence that patients with autism who participate in EAAT programs have substantially improved outcomes in their motor and sensory functioning.

### 5.7. Cognitive and Executive Functions

Although social cognition and executive functioning are not regularly examined features of ASD, in two articles in this review, the authors employed three different scales to evaluate the effects of the EAAT program on individuals with autism in these two domains: (1) the Kaufman Assessment Battery for Children II (K-ABC-II) (*n* = 1) [55]; (2) the Bayley Scales of Infant Development II (BSID-II) (*n* = 1) [55]; (3) the Tower of London (TOL) (*n* = 1) [61]. In a 2019 study [55], the authors indicated that the cognitive domains of the THR group substantially improved following an 8-week intervention compared with those preintervention; however, there was no statistically significant difference between the THR group and control group receiving conventional therapy. In a 2016 study [61], the authors found that the planning time for the problem-solving test decreased following the intervention, which likely indicated that EAT can improve the executive function in ASD patients. However, there was insufficient evidence from the two studies to conclude that EAAT training can substantially enhance the social cognitive and executive domains.

### 5.8. Family Functioning

In the previous literature, the authors highlight the importance of parental engagement and improved caregiver–child connections to enhancing the intervention outcomes [73], and consequently, family functioning is an essential indicator for assessing ASD [74]. In this review, in four out of twenty-five studies, the authors reported the family function outcomes using two different standardized assessments: (1) the Parenting Stress Index—Short Form (PSI-SF) (*n* = 1) [25] and (2) the McMaster Family Assessment Device (FAD) (*n* = 1) [54]. In two previous qualitative studies, the authors demonstrated that the EAAT programs were beneficial to family activities for children with autism, resulting in reduced parental stress [53,58]. In a 2020 study [54], the authors confirm the prior finding and offer preliminary evidence that EAAT with children with ASD has considerable benefits for family functioning; however, in a 2021 study [25], the authors suggest that EAAT did not reduce the parent distress after the intervention. Reports on the effects of EAAT programs on the families or parents of people with autism are still rare. While in three articles the authors preliminarily suggest that parental stress may be reduced by EAAT programs due to their therapeutic efficacy, in another article, the authors suggest the opposite. As a result, this evidence was limited, and we hope that researchers will quantitatively examine this crucial factor in future studies.

### 5.9. Other Skills

In addition to the above indicators, in eight studies, the authors report other helpful metrics that were hard to attribute to the above subdomains but nevertheless provided reliable qualitative or quantitative results. In four studies [37,53,58,59], the authors used laboratory methods that assessed the participant interaction and physiological state, as well as saliva or hair collection to measure the amount of cortisol present, with three out of the four studies providing data from baseline and after the intervention. According to the analysis, the data from the existing studies support that EAAT programs can significantly reduce the cortisol levels in participants (SMD = 0.53, 95% CI [0.02, 1.05], *p* = 0.04) (see Figure 9 for details). However, in a 2017 study [59], the authors report that there was no evidence to indicate a substantial improvement after the intervention. In conclusion, due to a lack of additional experimental and compared evidence, the veracity and reliability of the conclusion that EAAT programs can substantially reduce the cortisol levels in participants with ASD need to be questioned.

The other assessment methods included semi structured interviews, such as the Canadian Occupational Performance Measure (COPM) (*n* = 1) [28] and Goal Attainment Scaling (GAS) (*n* = 1) [43], which revealed that, after receiving EAATs, the participants were more effective at achieving their stated performance goals. The result was similar to that of a previous 2015 study [62], in which the authors used the Pedagogical Analysis and Curriculum (PAC) test. Additionally, the authors used the empathizing–systemizing quotients (EQ/SQ) (*n* = 1) [60], Ariel Performance Analysis System (APAS) (*n* = 1) [62], and Pediatric Quality of Life (PedsQL) 4.0 generic score scales (*n* = 1) [42] to evaluate the different aspects of the effects after the EAAT programs. In a 2018 study, the authors noted the advantages of ecopsychology following EAI for both the parents and children [58].

### 5.10. Persistence of Intervention Effect

Although in many studies, the authors report the considerable intervention effects of EAAT in many areas for individuals with autism, in only a few trials have the authors examined whether these effects were sustained. The sustainability of the intervention effects is a key factor in assessing the overall effectiveness of the intervention. In one study [38], the authors reveal that EAATs led to short-term improvements; however, the children’s behavior returned to baseline once the intervention ended. However, in another study [75], the authors discovered sustained behavioral gains even after the equine intervention was completed. Overall, the inconsistent information on the long-term benefits of EAATs makes it difficult to draw solid conclusions about the persistence of the therapeutic effects after interventions in ASD.

## 6. Discussion

### Summary of Results

The purpose of conducting this review was to use systematic review and meta-analysis techniques to evaluate the effectiveness of EAAT as a supplementary treatment for individuals with ASD. In our review, we combined qualitative and quantitative methodologies to assess two main aspects: (1) the social and communication functioning and (2) self-regulation functioning, and we objectively assessed the effectiveness of EAAT as an adjunctive therapy for ASD.

We selected 25 articles from 382, and of the 25 included studies, sixteen trials included control groups, and nine trials did not. For the study quality, we used the Sackett level of evidence, PEDro, and Cochrane’s “Risk of Bias Table”. According to the results, 11 of the 25 studies had qualities of Ⅰ and Ⅱ, only one study had a quality of Ⅴ, and the remaining 13 studies had qualities of Ⅲ and Ⅳ. Seven of the eight RCTs included in this study had scores of 6 or above, and one study had a score of 5. However, of all the included trials, we only assessed ten studies that were blinded or partially blinded, and in future trials, researchers should be blinded to ensure the accuracy of the measurements.

For most of the trials in all 25 included articles, the authors report the EAAT effects on multiple subsystems in ASD. As shown in Table 7, the social and behavioral domains were the most assessed for ASD, and while it appeared from the review that the EAAT programs substantially improved the social and behavioral skills of the participants with autism, some researchers reached the opposite conclusion, especially in terms of the subdomains, such as social awareness, social mannerisms, and social motivation, for which the meta-analysis revealed no substantial improvements. Moreover, there were no substantial improvements in the lethargy and stereotypes subdomains. The possible reasons for this were bias due to the length of the treatment, differences in the sites, or small samples. Although we did not have sufficient evidence on the program duration that provides the best treatment outcome for participants, according to a 2020 study [28], adolescents need weeks or months to learn new skills and change their behaviors, which suggests that interventions should be longer than 5 weeks. Similarly, according to the meta-analyses, EAAT programs substantially improve the skills in individuals with autism; however, in only two studies do the authors provide enough raw data that researchers should consider the EAAT program effects on language skills in individuals with autism in subsequent studies whenever possible. In the remaining four meta-analyses, because the authors did not set up control groups or obtain data from control groups in the original experiments, although their results indicate that EAAT programs substantially improve the motor and perceptual function and reduce the cortisol levels in patients with autism, this conclusion needs to be supported by future trials in which the researchers include control group data.

## 7. Conclusions

In this review, we utilized systematic review and meta-analysis methodologies to explore the findings of 25 studies on the impact of EAAT programs on individuals with ASD. The included studies provided us with sufficient data to draw the conclusion that EAAT programs substantially improve the social and behavioral functions in people with ASD, which is broadly in line with other research findings. The results also indicate substantial improvements in the language abilities and motor and sensory functioning. However, in only 11 studies did the authors achieve a quality rating of Ⅱ or higher, while the qualities of the remaining 14 studies still need to be improved, especially in terms of the following: the lack of control conditions; small sample sizes; unknown inclusion criteria; inability to randomly assign experimental and control groups; inability to set up control groups; dependence on parental assessment, single-assessment methods, etc. In addition, the effect assessment of the study should be stretched over a longer period, considering whether various aspects of the participants’ functioning are maintained after a considerable period. We hope that researchers will provide more evidence in the future.

## Figures and Tables

**Figure 1 ijerph-20-02630-f001:**
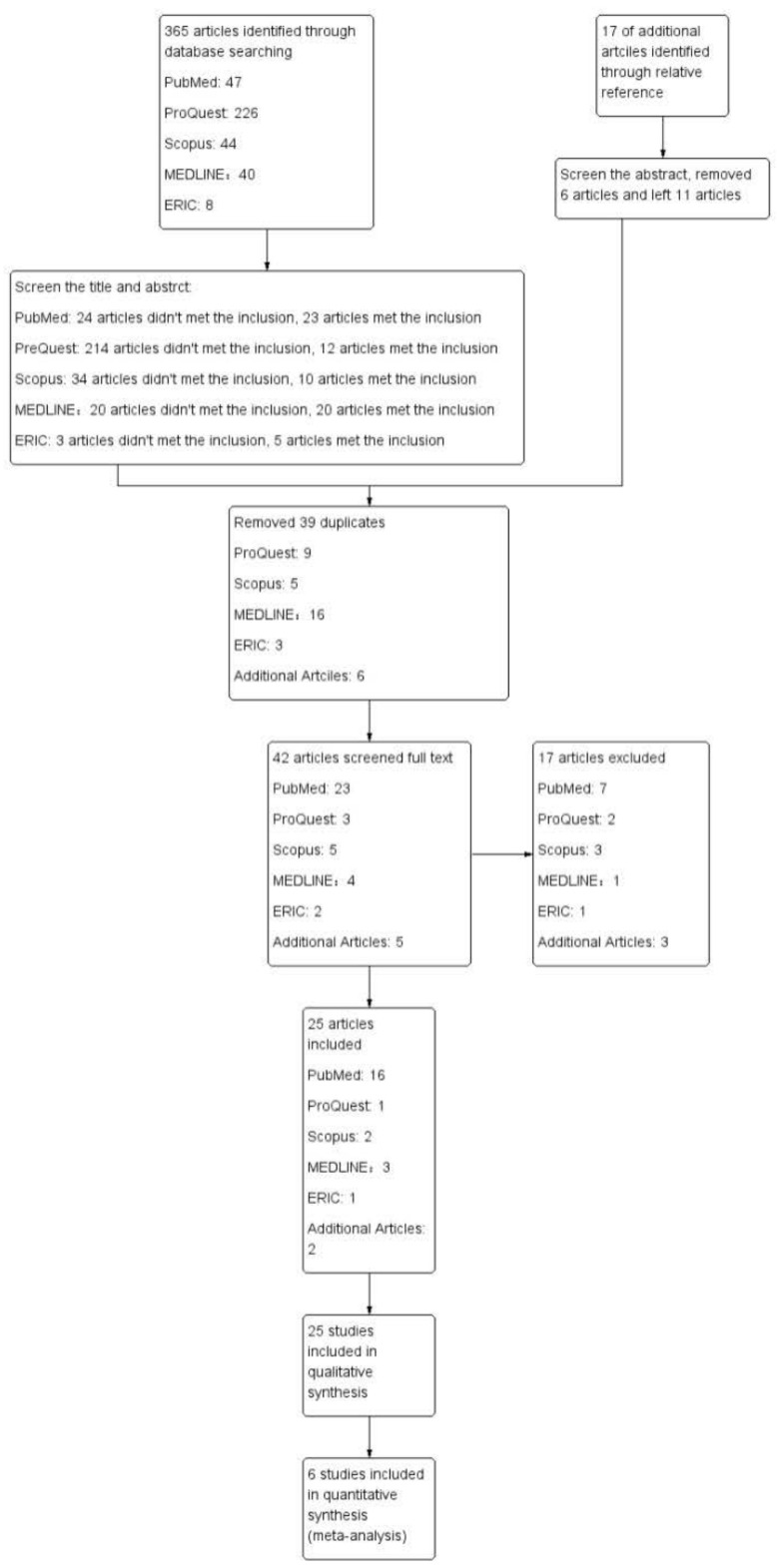
Flow diagram.

**Figure 2 ijerph-20-02630-f002:**
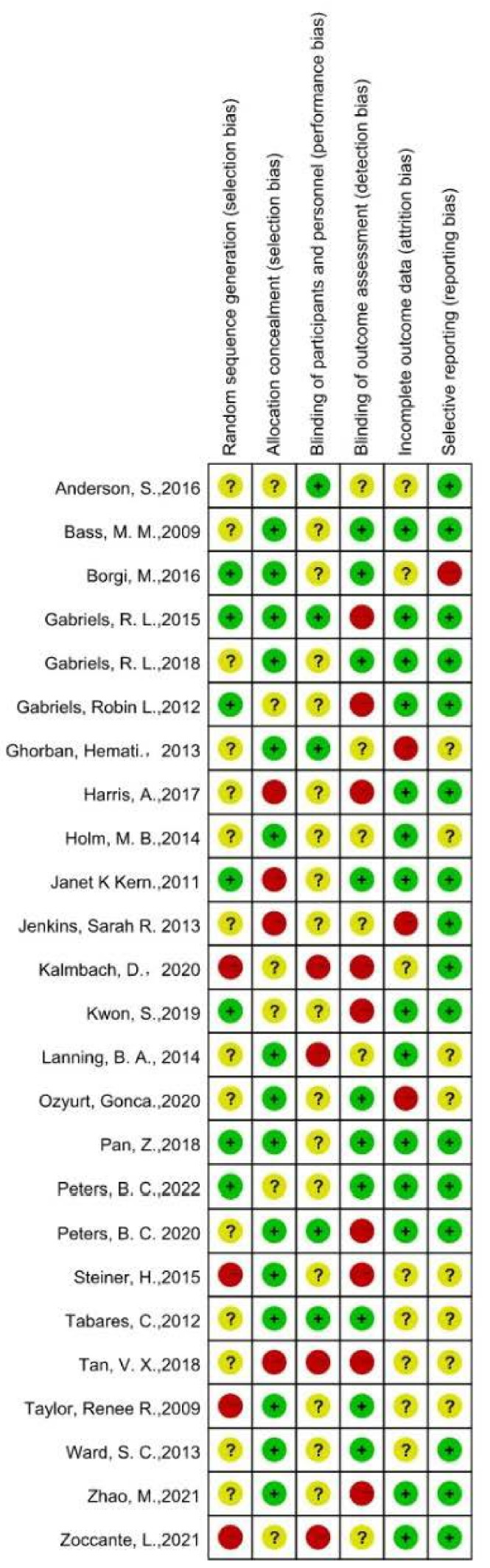
Risk of bias summary [25,28,30,38,40,42,43,53,54,55,56,57,58,59,60,61,62,63,64,65,66,67,68,69,70].

**Figure 4 ijerph-20-02630-f004:**
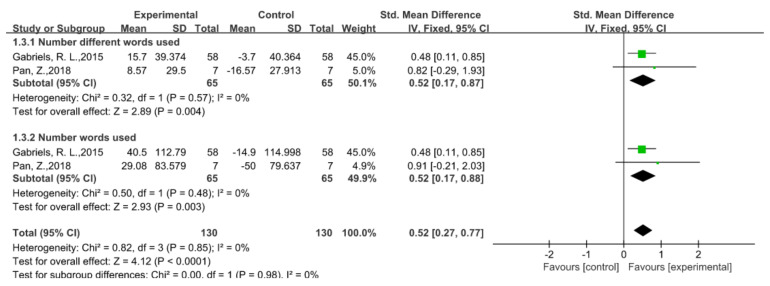
Forest plot of language ability using SALT [30,56].

**Figure 5 ijerph-20-02630-f005:**
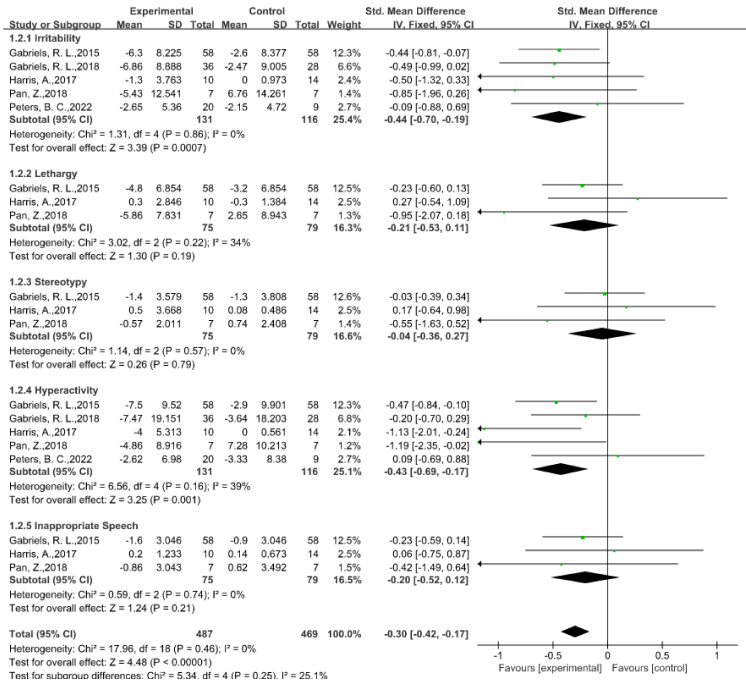
Forest plot of behavioral function using ABC-C [30,43,56,57,59].

**Figure 6 ijerph-20-02630-f006:**
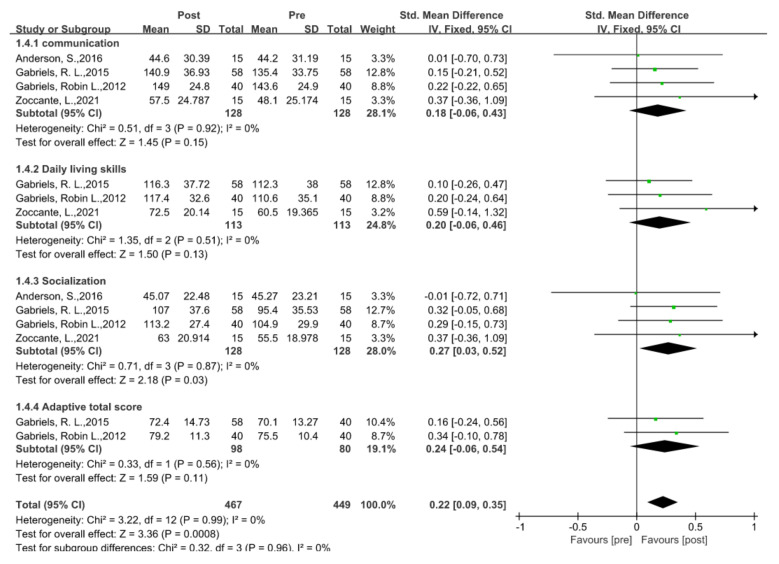
Forest plot of behavioral functioning (pre-post design) using VABS [25,30,60,66].

**Figure 7 ijerph-20-02630-f007:**
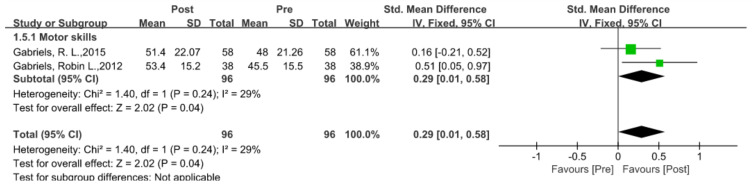
Forest plot of motor function (pre-post comparison) using BOT-2 [30,66].

**Figure 8 ijerph-20-02630-f008:**
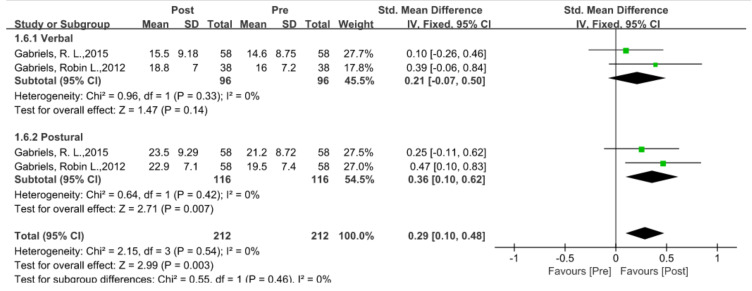
Forest plot of sensory function (pre-post comparison) using SIPT [30,66].

**Figure 9 ijerph-20-02630-f009:**
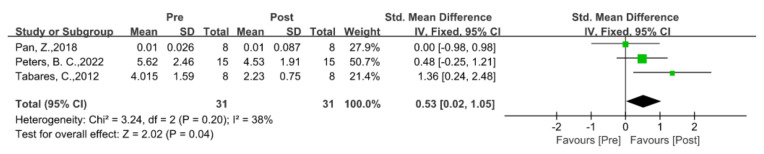
Forest plot of cortisol levels [43,56,67].

**Table 1 ijerph-20-02630-t001:** Inclusion and exclusion criteria.

Inclusion Criteria
(1) Article published in a journal with peer review;
(2) Publication in English language;
(3) Investigates the use of experimental or quasi-experimental designs in the reporting of the outcome data for the “EAATs” of “ASD”.
**Exclusion Criteria**
(1) Publication in languages other than English;
(2) Literature reviews (including systematic reviews), commentaries, etc.;
(3) Animal-assisted or pet-assisted therapies for ASD that used animals other than horses;
(4) No clear outcomes.

## Data Availability

The data can be directed to the corresponding author.

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
