# Peer review of "Effects of Equine-Assisted Activities and Therapies for Individuals with Autism Spectrum Disorder: Systematic Review and Meta-Analysis"

_ijerph, 2023, doi:10.3390/ijerph20032630_

Round 1

Reviewer 1 Report

Thank you for the opportunity to review this manuscript. This is a very important subject and I thoroughly enjoyed reading it- thank you.

Title - good description of the study - Effects of Equine-Assisted Activities and Therapies for Individuals with Autism Spectrum Disorder: Systematic Review and Meta-Analysis.

Relevance of the manuscript

The article is within the scope of International Journal of Environmental Research and Public Health. Enlarged knowledge about the Effects of Equine-Assisted Activities and Therapies for Individuals with Autism Spectrum Disorder is much needed.

 References are relevant and up to date.

The aim of the study is clearly described. The description of the design and methods of the study is clearly described and easy to grasp by the reader.

The content in the result is interesting, relevant, and presented in a structured and logical manner.

The conclusions of this study summarize the fact that the included studies provide sufficient data to draw the conclusion that EAAT programs substantially improve the social and behavioral functions as well as language abilities and motor and sensory functioning in people with ASD, which is broadly in line with other research findings. The limitations of some of the included studies are the lack of control conditions such as small sample sizes; unknown inclusion criteria; inability to randomly assign experimental and control groups; inability to set up control groups; dependence on parental assessment, single-assessment methods, etc. In addition, the effect assessment of the studies should be stretched over a longer period to be able to provide more evidence considering whether various aspects of the participants’ functioning are maintained after a consider able period.

Author Response

Dear reviewers,

We feel great thanks for your professional review work on our manuscript and gave us constructive comments. We have carefully considered your valuable suggestions and reread the entire paper.

As you, we expect further studies that will offer enough convincing evidence to back up the effectiveness of equine-assisted therapy for autism and investigate its underlying causes.

Once again, we appreciate your valuable advice despite your busy schedule and very glad that you are interested in our manuscript.

Reviewer 2 Report

The work presented in this paper is interesting based on a  power of EAATs treatment effect on people with ASD that is still not so common (the authors analysed findings of 25 studies). The manuscript is a systematic review that aims to evaluate a number of aspects that influence how well the EAAT treatments  work on people with ASD.
General comments: The equine-assisted therapies such as therapeutic horseback riding (THR) and hippotherapy (HT) - are exercise therapies that can have positive physical effects on coordination, muscle tone, postural alignment, stiffness/flexibility, endurance and strength, correcting abnormal movement patterns and improving gait and balance but on the other hand this unique therapies or interventions including horses are known to have a positive impact on  cognitive and emotional functions as well as prosocial health. This multidimensional effect can be very important as an alternative method of ASD treatment and this is why the studies on a power of EAATs applied for people with ASD are practically significant. Detailed analysis of studies on ASD treatment can suggest that there is still need to improve the designs and methodologies of these researches. The work is generally good, although some minor revisions (in the Introduction section) can be done to improve the quality of the paper and attract a wider readership. Suggested changes are included in the specific comments.
Specific comments: The introduction is generally fine, but it could be improved. In the section 1.2. Therapies for ASD: it could be mentioned that as ASD is a multifactorial desease  different types of treatment can be applied and Complementary and Alternative Methods (CAM) may support the classic medical approach (Volkmar et al 2014, Frye and Rossignol 2016, Zoccante et al 2021). Swimming, art therapy, music therapy and equine-assisted activities and therapies (EAATs) are metioned among the most implemented and effective types of CAM (Zoccante et al.2021). In lines 95 and 96 such information (about several species used in animal-assisted interventions) should be mentioned in the previous section (1.3. Animal-Assisted Interventions (AAIs) for ASD), not in the section 1.4. EAATs for ASD. On the other side, equine-assisted interventions are used in therapy and rehabilitation of people with various types of disabilities (not only autism, but also e.g. cerebral palsy, intellectual disabilities, MS, PTSD) and this information could be included in the section 1.4. The authors mentioned that the EAATs benefits are different from those of other animal-assisted therapies and it would be good to underline positive impact of regular hippotherapeutic sessions on both, physical and mental functions in more detailed way. Moreover, in the section 1.4. more detailed information could be added to explain what are particular riding and nonriding activities (mentioned in line 121) that are used especially for people with ASD. Also equine-facilitated learning (EFL) could be described with more details as it is significantly different from those of other forms of horse-assisted activities.

Author Response

Dear reviewers,

Thank you for your constructive comments on our manuscript. We have carefully considered your valuable suggestions, and all the concerns raised have been addressed in detail as follows, point-to-point:

Point 1: In the section 1.2. Therapies for ASD: it could be mentioned that as ASD is a multifactorial desease  different types of treatment can be applied and Complementary and Alternative Methods (CAM) may support the classic medical approach (Volkmar et al 2014, Frye and Rossignol 2016, Zoccante et al 2021). Swimming, art therapy, music therapy and equine-assisted activities and therapies (EAATs) are metioned among the most implemented and effective types of CAM (Zoccante et al.2021).

Response 1: In the section 1.2. Therapies for ASD, lines 86–90, we have carefully taken into account your suggestions and added the following: Meanwhile, since ASD is a multifactorial disease, numerous treatment options have become available, and complementary and alternative medicine (CAM) may increasingly be used alongside classical medical practices to treat ASD (Volkmar et al 2014, Frye and Rossignol 2016, Zoccante et al 2021). Among the most popular and successful forms of CAM were swimming, music therapy, art therapy, and animal-assisted interventions (AAIs)  (Zoccante et al 2021) .

Point 2: In lines 95 and 96 such information (about several species used in animal-assisted interventions) should be mentioned in the previous section (1.3. Animal-Assisted Interventions (AAIs) for ASD), not in the section 1.4. EAATs for ASD.

Response 2: Thank you for your valuable suggestions. We have moved lines 95,96 in the section 1.4. EAATs for ASD (AAIs include a variety of animals, such as dogs, horses, rabbits, dolphins, guinea pigs, and llamas) to lines 94-95 (in the section 1.3. Animal-Assisted Interventions) in order to make the structure of the article more logical in the context.

Point 3:  On the other side, equine-assisted interventions are used in therapy and rehabilitation of people with various types of disabilities (not only autism, but also e.g. cerebral palsy, intellectual disabilities, MS, PTSD) and this information could be included in the section 1.4. 

Response 3:  Thank you for your reasonable suggestion. After considering your suggestion, we added "EAATs can help people with autism, cerebral palsy, intellectual disabilities, multiple sclerosis (MS), and post-traumatic stress disorder (PTSD), among other conditions (Pálsdóttir, Anna M, 2020; Megan Kiely Mueller, 2017). And of all the animal-assisted therapies for ASD, the EAAT program is the most widely utilized (O'Haire, M, 2017)”content in lines 103–106 (in the section 1.4. EAATs for ASD) and added two reasonable citations to make sure the article is more attractive.

Point 4:  The authors mentioned that the EAATs benefits are different from those of other animal-assisted therapies and it would be good to underline positive impact of regular hippotherapeutic sessions on both, physical and mental functions in more detailed way.

Response 4:  Thank you for your suggestion. We have made the following changes in response to your suggestion: ”Another analysis revealed that the hippotherapy (HIP) exercises had a beneficial effect on postural control, interpersonal relationships, and adaptive behaviors (Phyllis & Erdman, 2015). Therefore, horses can offer people with ASD a special way of fostering positive social engagement" (in lines 113–115, 1.4, EAATs for ASD).

Point 5:  Moreover, in the section 1.4. more detailed information could be added to explain what are particular riding and nonriding activities (mentioned in line 121) that are used especially for people with ASD.

Response 5:  Thank you for your suggestion. For the sake of the integrity of the article, we have made the following changes after careful consideration: “The fundamental and core idea behind THR is to engage people with ASD in horseback riding and nonriding activities (Barn activities, such as cleaning the barn, feeding horses, and watching the horses' motions) with licensed instructors, counselors, or equestrians who teach them horsemanship skills that target improving their physical, behavioral, and prosocial health” (in lines 132-133, section 1.4. EAATs for ASD)

Point 6:  Also equine-facilitated learning (EFL) could be described with more details as it is significantly different from those of other forms of horse-assisted activities.

Response 6: Thanks for your suggestion. We made changes as follows: “Notably, as opposed to EATs and HIP, equine-facilitated learning (EFL) is a distinct experiential learning technique that blends learning abilities and interaction with horses (ponies, miniature horses, donkeys, and mules) with individual therapy and emotional regulation to strengthen children's awareness and control of their emotions, cognition, and behavior (Pendry, Patricia, 2014).” (in lines 139-143, section EAATs for ASD).

Reviewer 3 Report

It is an important revue but qualitative results, even if interesting for a general view of te research, have no scientif soundness and it would be clearly represented

The idea of verifying the efficacy of intervention measuring the parents mental health could be interesting too, but it needs a specific research and pheraps you will have to consider a lot of bias, firt of all the definition of "mental health"

Author Response

Dear reviewers,

We feel great thanks for your professional review work on our manuscript and gave us constructive comments. We have carefully considered your valuable suggestions and reread the entire paper and share your expectation for more robust experimental evidence of the specific benefits of horse-assisted therapy for autism, as well as exploring the mechanisms behind this.

Similarly, we believe that parental mental health status is a risk factor for autism education, but as you said, we still need to investigate how to define "mental health status" and how to design such research. We hope we can explore this factor in the future.

Once again, we appreciate your valuable advice despite your busy schedule.
